# AFD-INSTRUCTION: A COMPREHENSIVE ANTIBODY INSTRUCTION DATASET WITH FUNCTIONAL ANNOTATIONS FOR LLM-BASED UNDERSTANDING AND DESIGN

**Ling Luo**[♣][*], **Wenbin Jiang**[♣][*], **Hongyuan Chang**[O], **Xinkang Wang**[♣], **Xushi Zhang**[♣],
**Yueting Xiong**[◇][♦][†], **Mengsha Tong**[♣][♠][†], **Rongshan Yu**[♣][♡][†]

[♣] National Institute for Data Science in Health and Medicine, Xiamen University
[O] Institute of Artificial Intelligence, Xiamen University
[♠] School of Life Sciences, Xiamen University
[♡] School of Informatics, Xiamen University
[◇] State Key Laboratory of Vaccines for Infectious Diseases, Xiamen University
[♦] Xiang An Biomedicine Laboratory
{luoling2001}@stu.xmu.edu.cn, {wenbin_jiang1}@163.com,
{mstong, ytxiong, rsyu}@xmu.edu.cn

## ABSTRACT

Large language models (LLMs) have significantly advanced protein representation learning. However, their capacity to interpret and design antibodies through natural language remains limited. To address this challenge, we present **AFD-Instruction**, the first large-scale instruction dataset with functional annotations tailored to antibodies. This dataset encompasses two key components: antibody understanding, which infers functional attributes directly from sequences, and antibody design, which enables de novo sequence generation under functional constraints. These components provide explicit sequence–function alignment and support antibody design guided by natural language instructions. Extensive instruction-tuning experiments on general-purpose LLMs demonstrate that AFD-Instruction consistently improves performance across diverse antibody-related tasks. By linking antibody sequences with textual descriptions of function, AFD-Instruction establishes a new foundation for advancing antibody modeling and accelerating therapeutic discovery.

## 1 INTRODUCTION

Large Language Models (LLMs), such as GPT (OpenAI, 2023), DeepSeek (Ren et al., 2025), Llama (Grattafiori et al., 2024), and Qwen (Yang & Team, 2025), have emerged as a powerful paradigm in Natural Language Processing (NLP). With billions of parameters, these models are trained on vast text corpora and excel at generating human-like text and understanding complex contexts. To unlock the potential of LLMs for deciphering proteins, researchers have turned to instruction tuning. For example, Mol-Instructions (Fang et al., 2024b) curates a biomolecular instruction dataset and benchmarks fine-tuned LLMs on protein-specific instruction-following tasks such as function description. InstructProtein (Wang et al., 2024b) leverages a structured knowledge graph to generate higher-quality protein–text instruction data. ProtLLM (Zhuo et al., 2024) further constructs InterPT, a large interleaved protein–text corpus including multi-protein prompts, to train instruction-following models that integrate protein information.

Antibodies represent a particularly valuable yet challenging class of proteins. As immunoglobulins that naturally evolve to recognize and bind specific antigens with varying affinity, they are crucial in

---

[*] Equal contribution and shared co-first authorship.
[†] Corresponding author.

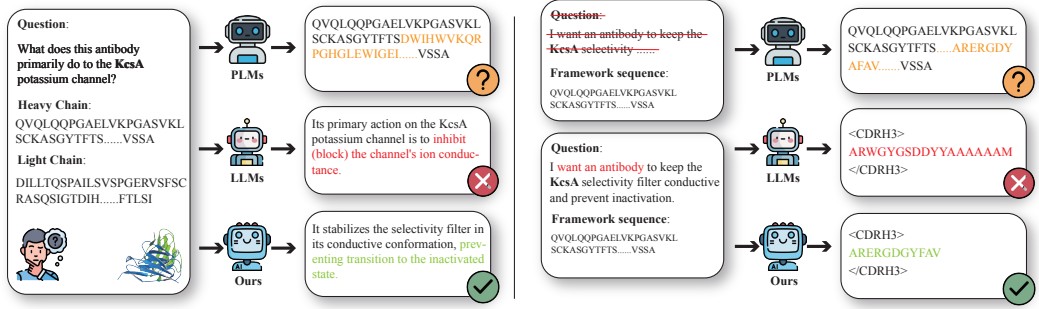

Figure 1: Traditional LLMs fail to interpret antibody sequences, while protein language models (PLMs) cannot understand natural language and thus cannot incorporate human functional descriptions into antibody design. Our proposed **AFD-Instruction** bridges this gap by aligning antibody sequence understanding with natural language guidance, enabling functional and design-oriented reasoning across both modalities.

the immune response. However, most existing Protein Language Models (PLMs) are trained in an unsupervised manner on raw sequences without integrating functional signals. While they capture evolutionary patterns, they often struggle with antibody-specific properties such as neutralization potency and precise antigen recognition. Previous research has raised concerns about whether evolutionary signals alone can optimize functions such as antigen binding (Hie et al., 2024). A major challenge is the lack of resources that align antibody sequences with experimentally validated functions. For example, the Observed Antibody Space (OAS) database (Kovaltsuk et al., 2018; Olsen et al., 2022) contains millions of sequences, but most lack annotations on targets or biological activities, resulting in underutilization of the available data. Recent studies have sought to introduce supervision by incorporating structural features into language models (Shanker et al., 2024; Barton et al., 2024) or training sequence-based predictors for antigen binding (Jing et al., 2024; Kalemati et al., 2024), yet a comprehensive resource linking arbitrary antibody sequences with detailed natural-language functional descriptors remains absent.

To address this gap, we introduce **AFD-Instruction**, a large-scale resource that aligns antibody sequences with concise, machine-readable functional descriptors. After reviewing approximately 4,000 articles through a multi-agent extraction pipeline with expert verification, the dataset comprises over 430K validated instructions. Each entry links a antibody sequence to a brief, normalized description—covering target antigen, binding specificity or affinity, and neutralization or blocking activity—formatted for instruction tuning. Our goal is to embed LLMs with antibody-specific functional annotations, enabling them to infer function from sequence and generate sequences that satisfy specified functional constraints. To assess effectiveness, we fine-tuned several LLMs on AFD-Instruction and evaluated them across two use cases: sequence understanding (inferring or explaining function from sequence) and generative design (targeted CDR3 design or de novo antibody generation). Across diverse architectures, AFD-Instruction-tuned models consistently outperformed sequence-only baselines in functional classification and target inference, while producing designs that more faithfully reflected desired functional profiles. These results demonstrate that explicit sequence–function supervision enhances interpretability and controllability, bridging the gap between abundant sequences and scarce functional annotations, and providing a practical foundation for transparent, goal-directed antibody discovery.

## 2 RELATED WORKS

**Protein-Language Datasets** Protein-language datasets have recently emerged to link protein sequences with textual annotations, facilitating the adaptation of LLMs to biological tasks. Early resources, such as UniProtKB/Swiss-Prot (Consortium, 2023), provide high-quality functional descriptions by experts but are limited in scale. To improve coverage, various augmentation strategies have been proposed. Notable examples include SwissProt-Aug (Yuan et al., 2024), which leverages LLMs to expand curated annotations, and Evola (Zhou et al., 2025), which generated hundreds of

millions of protein-related question-answer pairs for instruction tuning. Other databases integrate ontological or domain-specific knowledge, such as ProteinKG25 (Zhang et al., 2022b), aligning sequences with Gene Ontology definitions and enzyme-focused corpora from KEGG/UniProt descriptions. (Hoarfrost et al., 2024). While these datasets establish the foundation for protein-language modeling, they are either general-purpose or domain-specific. None specifically target antibodies. Due to the specificity and scarcity of data, establishing systematic resources for antibodies remains a significant challenge. A comparison of representative datasets is provided in Table 5.

**Large Language Models** The advancement of LLMs represents a significant milestone in natural language processing, demonstrating exceptional performance across various complex tasks (Devlin et al., 2018; Longpre et al., 2023; Chowdhery et al., 2022; OpenAI et al., 2024; Anthropic, 2024). Beyond linguistic applications, these models are increasingly applied to diverse scientific domains, including mathematical problem solving (Wei et al., 2022; Imani et al., 2023), physics simulation (Ali-Dib & Menou, 2023; Cherian et al., 2024), drug discovery (Liang et al., 2023; Liu et al., 2023), materials discovery (Jia et al., 2024; Gan et al., 2025), and other emerging scientific fields. Recently, LLMs have also been adapted for protein studies, showing promise in capturing sequence patterns and aligning protein-language representations (Wang et al., 2024a; Fang et al., 2024a). However, the scarcity of antibody-specific datasets remains a critical barrier to progress. This limitation impedes the comprehension and design of antibody sequences facilitated by LLMs.

**Protein-Text Modeling** The initial attempt by Galactica (Taylor et al., 2022) involved pretraining an LM on text with only limited protein sequences, leading to weaker protein comprehension compared to protein language models (PLMs). To address this, subsequent efforts (Xu et al., 2023; Liu et al., 2024a) integrated PLMs and LMs through cross-modal contrastive learning (Radford et al., 2021). However, such alignment remains inadequate for protein-to-text generation, which requires conditional generation grounded in protein understanding rather than simple representation matching. Recently, ProteinChat (Guo et al., 2023) introduced a linear projector that maps PLM embeddings to an LM to enhance its awareness of proteins. Concurrently, ProtT3 (Liu et al., 2024b) directly tackled the challenge of protein-to-text generation; ProtLLM (Zhuo et al., 2024) proposed an interleaved pretraining paradigm treating proteins as words. InstructProtein (Wang et al., 2024a) aligned protein and natural language representations through instruction tuning, further advancing the integration of protein and language. Despite the considerable biomedical relevance of antibodies, progress has been impeded by the lack of antibody-specific data necessary to develop specialized models. The AFD-Instruction provides a foundational resource for enabling LLMs to interpret antibody sequences while utilizing natural language as guidance for antibody design.

## 3 AFD-INSTRUCTION CONSTRUCTION

We introduce AFD-Instruction, a large-scale antibody instruction dataset with functional annotations, which serves as a foundational resource for LLMs to understand antibody sequence functionality and facilitate antibody design, as illustrated in Figure 2 (C). To construct AFD-Instruction, we first collected antibodies from SabDab (Raybould et al., 2020) and the Protein Data Bank (PDB) (Berman et al., 2000), focusing on those commonly used for antibody design. To mitigate data imbalance (Wang et al., 2024a), we employed *MMseqs2* (Steinegger & Söding, 2017) to compute pairwise sequence distances. Specifically, let $A = a_1 a_2 \ldots a_n$ and $B = b_1 b_2 \ldots b_m$ denote two antibody sequences of lengths $n$ and $m$, respectively. Before calculating distances, we define a substitution matrix to obtain the replacement score $s(\cdot, \cdot) \in (0, 1]$, along with a gap penalty scheme $W_k$, where $k$ denotes the length of the gap. The dynamic programming distance matrix $H$ is then formulated as:

$$H_{i,j} = \min \{H_{i-1,j-1} + s(a_i, b_j), \ H_{i-k,j} - W_k, \ H_{i,j-1} - W_1, \ 1\}, \tag{1}$$

with boundary conditions $H_{k,0} = H_{0,l} = 0$ for $0 \leq k \leq n$ and $0 \leq l \leq m$. The final normalized sequence distance between $A$ and $B$ is given by:

$$d_s(A, B) = \frac{H_{n,m}}{\max(n, m)}. \tag{2}$$

Based on this metric, we sampled a balanced set of 4,305 antibody entries. For each antibody, we then retrieved the corresponding publication to obtain the textual functional descriptions that form

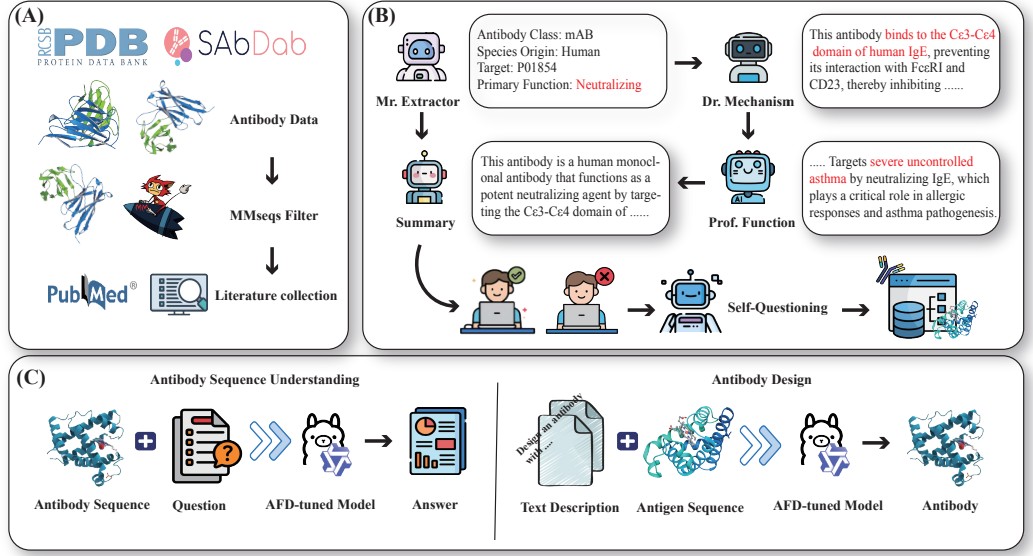

Figure 2: Overview of AFD-Instruction. (A) Antibody data and related literature are collected from PDB/SAbDab and PubMed and filtered. (B) presents the dataset construction pipeline. (C) highlights two downstream tasks enabled by AFD-Instruction, namely antibody sequence understanding and antibody design guided by functional annotations.

the core annotations of the dataset. The overall data construction pipeline is illustrated in Figure 2 (B). Building upon this foundation, we further design a multi-agent workflow to coordinate key subtasks, including data extraction, quality verification, and semantic alignment, in order to ensure the generation of high-fidelity antibody–description pairs.

## 3.1 MULTI-AGENT SYSTEM

To enhance clarity and robustness, the analysis pipeline is structured as a multi-agent system in which each agent assumes a distinct and complementary role. At the entry point is **Mr. Extractor**, responsible for scanning the text to gather essential antibody-related information, including class, target, origin, and primary function. This agent ensures that the initial description is faithfully captured, thus providing a reliable foundation for subsequent refinement. Once the groundwork is established, **Dr. Mechanism** takes over. With an analytical perspective, this agent examines the structural and mechanistic aspects, tracing how the antibody binds, where the interaction occurs, and what molecular and biological effects are triggered. The task here is to enrich the initial record with mechanistic depth and precision. Finally, **Prof. Function** offers the higher-level interpretation, including synthesizing the structural and mechanistic details into a coherent account of the mode of action of the antibody, its therapeutic relevance, and the supporting evidence. Beyond summarization, it will also highlight the unique features of this antibody and its importance in medicine or clinical practice.

## 3.2 SELF-QUESTIONING

**Antibody Sequence Understanding**     To expand AFD-Instruction from descriptive annotations to task-oriented supervision, we adopt a *self-questioning* strategy (Sachan & Xing, 2018; Chen et al., 2025) to automatically create instruction–response pairs. The generated tasks cover both categorical and open-ended settings. Classification-style instructions require the model to make discrete judgments, such as identifying the structural class of an antibody, determining whether it possesses a particular function, or assessing its therapeutic applicability to a given disease. In contrast, non-classification tasks focus on interpretive and generative reasoning, prompting the model to explain the functional role of a given sequence, describe its binding mechanism, or infer its potential therapeutic context. To ensure clarity and reproducibility, antibody sequences are marked with explicit

chain delimiters: heavy chains use `<H></H>` tags and light chains use `<L></L>` tags. Starting from the antibody–description pairs curated by the multi-agent workflow, seed prompts are constructed and passed to a large language model to obtain diverse instructions. The outputs are subsequently subjected to automatic consistency checks and redundancy reduction processes, resulting in a large-scale instruction dataset. This instruction enables models not only to classify antibodies but also to reason about, compare, and interpret them within a unified framework.

**Function-Guided Antibody Design**    Beyond sequence understanding, we further extend the self-questioning paradigm to the task of function-guided antibody design. In this context, the input is no longer a fixed antibody sequence, but rather a specification of the desired functional property together with the corresponding antigen sequence, explicitly delimited by `<Anti></Anti>` tags. The output can be either a complete antibody sequence or, in the focused variant, a complementary determining region (CDR3) sequence, where the latter is explicitly enclosed by `<CDR3></CDR3>` tags. The objective is to generate either a complete antibody sequence or, in a more focused variant, a CDR3 sequence that satisfies the given functional requirements. In addition to self-questioning, we also employ a *template-based conversion* strategy, where structured metadata extracted from publications or databases is directly transformed into instruction–response pairs via predefined templates, ensuring faithful preservation of factual content. To ensure biological plausibility, generations that deviate from the intended functionality or exhibit structural inconsistencies are filtered out. This process allows the design-oriented dataset to enhance the understanding tasks, equipping large language models with the capacity not only to analyze antibody properties but also to conditionally generate novel candidates guided by explicit functional intent.

## 3.3 QUALITY CONTROL

To ensure reliability, we adopt a quality control pipeline combining automated checks with expert review. Antibody sequences from SabDab and PDB undergo integrity and duplication checks, followed by manual validation of 10% through random sampling. For instruction–response pairs, we perform automatic semantic checks, with two independent experts reviewing a 5% subset. The inter-annotator agreement measured by Cohen's $\kappa$ reached 0.82, indicating high consistency. When discrepancies arise, reviewers discuss and reconcile them. We then update the relevant templates or extraction rules to prevent recurrence. This iterative loop ensures that AFD-Instruction remains reliable, biologically grounded, and function-oriented for downstream modeling and design.

## 4 A CLOSER LOOK AT AFD-INSTRUCTION

### 4.1 TAXONOMY AND APPLICATION SCOPE OF INSTRUCTION TASKS

As illustrated in Figure 3 (a), AFD-Instruction is anchored around two core domains: antibody understanding, which includes classification-based Q&A tasks and open-ended caption tasks, and antibody design, which focuses on de novo design of the variable regions and targeted CDR3 design. Each domain's instruction tasks address essential challenges in antibody research, aiming to advance the field through the training and utilization of LLMs.

**Antibody Sequence Understanding**    Q&A tasks span five categories: antibody class, binding site, mechanism of action, disease association, and antibody function. In contrast, the caption tasks provide open-ended descriptions of antibody properties and functions, explicitly excluding antibody class. Together, these tasks form a foundation for understanding antibody recognition and functionality, which is critical for therapeutic antibody development, clinical translation, and drug discovery.

**Antibody Design**    These tasks address sequence generation under functional constraints, covering de novo design of variable regions and targeted design of CDR3 loops. The primary objective is to enable natural language–guided antibody design, thereby enhancing sequence diversity, refining binding specificity, and improving therapeutic potential. Such capabilities are essential for accelerating discovery pipelines and advancing next-generation biotherapeutics.

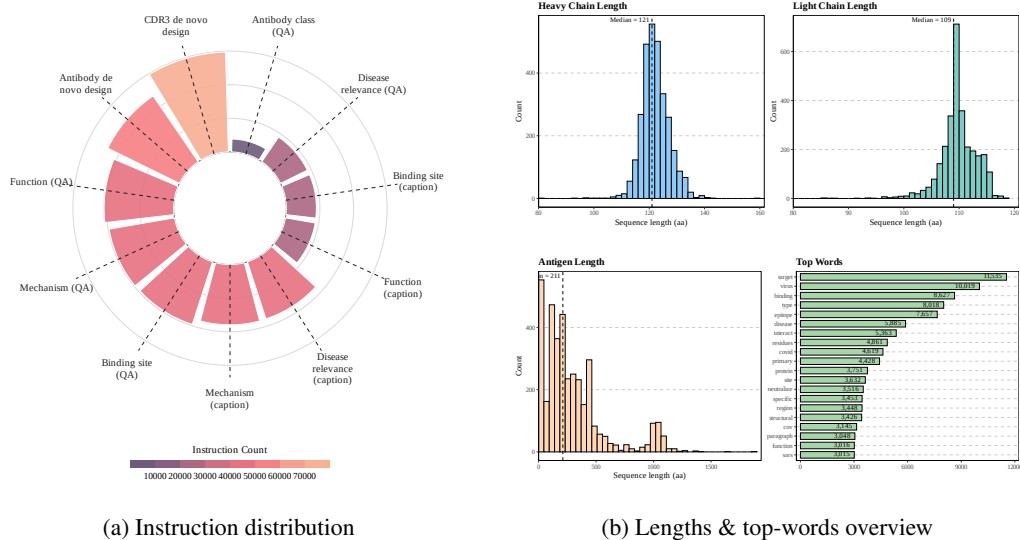

(a) Instruction distribution

(b) Lengths & top-words overview

Figure 3: Dataset overview: (a) rose plot of instruction counts; (b) combined lengths and word analysis.

## 4.2 DATASET DISTRIBUTION AND HETEROGENEITY

Figure 3 (b) illustrates the distributions of the heavy-chain, light-chain, and antigen sequence lengths in antibodies. The heavy-chain variable regions display a near-unimodal, bell-shaped distribution centered around 110–125 amino acids, reflecting the conserved FR1–FR4 framework architecture of immunoglobulins (Chothia & Lesk, 1987), while longer sequences indicate variability in the CDR-H3 region. Light-chain variable regions are generally shorter and exhibit less variability due to structural constraints of $\kappa/\lambda$ chains and typically shorter CDR-L3, with a pronounced peak indicating limited variation outside complementarity-determining regions. In contrast, antigen sequences display a broader, multimodal distribution: shorter sequences correspond to peptide epitopes; intermediate peaks relate to single domains; and longer sequences represent full-length proteins or viral polyproteins. The right-skewed tail results from including intact proteins alongside smaller fragments. Complementing the sequence-level statistics, the bottom-right panel summarizes word frequencies from a cleaned, stemmed bag-of-words pipeline that removes stop words and merges near duplicates. The corpus focuses on mechanistic/structural terms (bind, epitope, residue, site, interact, affinity, neutralize) and target-focused nouns (virus, protein, region). Subsets show task-aligned emphases: the binding-site subset favors contact/site descriptors; the mechanism subset favors action verbs and effect terms; the function and disease-relevance subset favors biological/clinical terminology and disease names (HIV-1, SARS-CoV-2, influenza). This breadth spans molecular to phenotypic descriptors and supports more mechanistically grounded representations. Appendix C provides further details about the dataset.

## 5 EXPLORING THE POTENTIAL OF AFD-INSTRUCTION

### 5.1 ANTIBODY SEQUENCE UNDERSTANDING

To assess whether AFD-Instruction enhances LLMs' understanding of antibodies, we perform instruction tuning across its five main domains using two widely adopted open-source foundation models, LLaMA-8B (Dubey et al., 2024) and Qwen2-7B (Yang et al., 2024). Furthermore, to investigate the robustness and scalability of AFD-Instruction across diverse parameter scales and model families, we also conduct instruction-tuning experiments on Gemma-9B (Gemma et al., 2025), DeepSeek-MoE-16B (Dai et al., 2024), and LLaMA-70B (Grattafiori et al., 2024), followed by further evaluations to assess their performance. Their performance is then compared against six domain-specific models (InstructProtein (Wang et al., 2024a), Mol-Instructions (Fang

et al., 2024a), Galactica (Taylor et al., 2022), BioT5 (Pei et al., 2023), ProtLLM (Zhuo et al., 2024), and ProtT3 (Liu et al., 2024b)), six general-purpose open-source LLMs (LLaMA, Qwen, OPT (Zhang et al., 2022a), Alpaca (Taori et al., 2023), Gemma-3 (Gemma et al., 2025), and DeepSeek-V3 (DeepSeek-AI et al., 2025)), and five closed-source baselines (Claude-3 (Anthropic, 2024), GPT-4o (OpenAI et al., 2024), o4-mini (OpenAI, 2025), Grok-3 (xAI, 2025), and Gemini-2.5 (Comanici et al., 2025)). We partition AFD-Instruction into training, validation, and test subsets, using the first two for instruction tuning and the last for evaluation. Detailed training procedures, extended results, and evaluation metrics are provided in Appendix E, G.1, and F.

Table 1: Performance comparison between AFD fine-tuned models and baseline large language models (LLMs) on antibody sequence understanding tasks in classification task. Top-performing results are highlighted in **bold**, while second-best performances are underlined.

| Model | Params. | Class | | Disease | | Binding | | Mechanism | | Function | |
|---|---|---|---|---|---|---|---|---|---|---|---|
| | | ACC | F1 | ACC | F1 | ACC | F1 | ACC | F1 | ACC | F1 |
| *Domain-Specific Models* | | | | | | | | | | | |
| Galactica | 1.3B | 47.78 | 32.69 | 25.34 | 20.22 | 50.52 | 35.33 | 59.90 | 59.04 | 54.74 | 48.49 |
| Mol-Instructions | 7.0B | 47.56 | 32.23 | 25.13 | 20.08 | 52.08 | 34.24 | 41.61 | 29.38 | 53.16 | 34.71 |
| BioT5 | 252M | 53.13 | 35.43 | 73.25 | 43.51 | 48.82 | 33.27 | 58.36 | 36.85 | 46.53 | 31.75 |
| InstructProtein | 1.3B | 52.44 | 34.40 | 74.66 | 42.75 | 47.91 | 32.39 | 58.36 | 36.85 | 48.51 | 35.02 |
| ProtLLM | 7.0B | 53.74 | 63.26 | 49.84 | 40.61 | 48.92 | 68.59 | 57.11 | 58.13 | 47.73 | 20.21 |
| ProtT3 | 1.3B | 44.20 | 31.46 | 45.90 | 36.27 | 40.24 | 37.36 | 57.50 | 57.20 | 32.02 | 32.01 |
| *Open-Source General-Purpose LLMs* | | | | | | | | | | | |
| OPT | 1.3B | 47.56 | 32.23 | 23.34 | 18.22 | 52.09 | 34.25 | 41.61 | 29.38 | 53.47 | 34.90 |
| Alpaca | 7.0B | 51.25 | 39.76 | 25.34 | 20.08 | 52.09 | 34.25 | 48.84 | 42.36 | 54.82 | 38.96 |
| gemma-3 | 4.0B | 49.35 | 40.59 | 13.15 | 13.09 | 52.30 | 38.80 | 44.83 | 37.26 | 54.53 | 51.09 |
| Qwen2 | 7.0B | 90.47 | 90.46 | 32.96 | 32.49 | 53.42 | 41.44 | 49.54 | 45.45 | 64.59 | 60.68 |
| LLaMA | 8.0B | 87.16 | 87.14 | 6.19 | 5.90 | 52.37 | 39.43 | 59.72 | 59.69 | 57.72 | 52.59 |
| DeepSeek-V3 | 671B | 93.99 | 93.95 | 74.45 | 42.68 | 47.88 | 34.01 | 59.20 | 43.67 | 49.39 | 45.76 |
| *Commercial Closed-Source LLMs* | | | | | | | | | | | |
| Claude-3 | – | 95.40 | 95.37 | 70.89 | 42.02 | 42.65 | 33.89 | 43.81 | 35.46 | 47.84 | 45.02 |
| GPT-4o | – | 82.02 | 81.83 | 72.15 | 48.89 | 50.31 | 46.59 | 63.99 | 58.39 | 56.17 | 55.80 |
| o4-mini | – | 89.98 | 89.75 | 50.19 | 33.62 | 15.25 | 13.84 | 23.49 | 20.34 | 50.08 | 44.75 |
| Grok-3 | – | 89.76 | 89.76 | 65.82 | 52.49 | 34.39 | 33.21 | 38.70 | 37.74 | 59.84 | 59.44 |
| gemini-2.5 | – | 92.15 | 92.04 | 26.07 | 24.22 | 9.86 | 9.58 | 12.55 | 12.02 | 54.38 | 53.46 |
| *Ours* | | | | | | | | | | | |
| **QwenAB** | 7.0B | **98.86** | 97.23 | 87.83 | 83.29 | 87.81 | 87.77 | 93.60 | 93.45 | 85.01 | 84.93 |
| **LLaMAB** | 8.0B | 98.48 | 96.23 | 85.11 | 80.91 | 87.01 | 86.96 | 92.91 | 92.76 | 83.81 | 83.74 |
| **GemmaAB** | 9.0B | 96.19 | 95.24 | **88.51** | **84.07** | **89.11** | **89.06** | **94.14** | **93.99** | **87.96** | **87.92** |
| **DeepSeekAB-MoE** | 16B | 98.70 | 97.89 | 86.59 | 79.35 | 88.24 | 88.22 | 94.01 | 93.85 | 86.73 | 86.71 |
| **LLaMAB-70B** | 70B | 98.59 | **98.23** | 87.11 | 82.25 | 88.35 | 88.29 | 93.69 | 93.53 | 87.45 | 87.41 |

As shown in Table 1, large-scale models such as GPT-4o and DeepSeek-V3 perform competitively on the basic task of antibody class identification, indicating that such knowledge is partially embedded during pre-training. However, for specialized tasks such as disease association, binding prediction, mechanism inference, and functional annotation, general-purpose LLMs fall short, highlighting their limited antibody-specific reasoning. Even domain-specific protein language models (e.g., Galactica, Mol-Instructions, InstructProtein), despite extensive training on protein data, struggle with antibody-focused tasks due to the gap between general representations and the detailed knowledge required. In contrast, our instruction-tuned models, **QwenAB** and **LLaMAB**, achieve state-of-the-art results across all subtasks. Specifically, **QwenAB** reaches 98.86% accuracy on class prediction and maintains superior performance in disease association (87.83% ACC, 83.29% F1) and functional annotation (85.01% ACC, 84.93% F1). On average, QwenAB improves accuracy by 20.21 points over the strongest baseline, while LLaMAB shows a 19.05 point gain, underscoring the effectiveness of AFD-Instruction in equipping foundation models with antibody-specific expertise.

In addition to classification benchmarks, we evaluate models on caption tasks that require generating free-form textual answers rather than discrete labels. Table 2 reports results for commercial closed-

source LLMs and a subset of open-source general-purpose models, while results for domain-specific and additional open-source baselines are provided in Appendix G.1. Interestingly, in these caption tasks, large general-purpose models substantially outperform domain-specific protein models, revealing a clear gap between protein-level modeling and antibody-specific language understanding. AFD-Instruction directly addresses this gap as the first antibody dataset with natural language functional annotations, markedly enhancing the antibody understanding capabilities of LLMs and delivering consistent improvements across all evaluated metrics compared to baseline models.

Table 2: Caption task antibody sequence understanding results. This table includes *Commercial Closed-Source LLMs* and a subset of *Open-Source General-Purpose LLMs*; extended results covering *Domain-Specific Models* and the remaining open-source baselines appear in the Appendix G.1. Top-performing results are highlighted in **bold**, while second-best performances are underlined.

| MODEL | EXACT↑ | BLEU-2↑ | BLEU-4↑ | ROUGE-1↑ | ROUGE-2↑ | ROUGE-L↑ | METEOR↑ |
|---|---|---|---|---|---|---|---|
| *Binding* | | | | | | | |
| Qwen2 | 0.00 | 10.62 | 4.60 | 25.78 | 10.17 | 24.04 | 23.39 |
| LLaMA | 0.00 | 8.49 | 3.36 | 23.67 | 7.17 | 22.47 | 15.93 |
| Claude-3 | 0.75 | 9.12 | 3.14 | 16.53 | 4.62 | 15.82 | 18.58 |
| GPT-4o | 1.40 | 14.36 | 6.74 | 22.92 | 7.83 | 22.04 | 23.50 |
| o4-mini | 0.43 | 12.51 | 4.68 | 22.06 | 7.76 | 20.74 | 19.51 |
| Grok-3 | 1.82 | 22.19 | 11.42 | 30.03 | 11.96 | 28.92 | 30.42 |
| **QwenAB** | **4.24** | 28.90 | **17.25** | 37.95 | **18.84** | 36.85 | 37.80 |
| **LLaMAB** | 4.10 | **29.78** | 17.05 | **38.49** | 18.68 | **37.33** | **38.84** |
| *Disease* | | | | | | | |
| Qwen2 | 0.23 | 22.88 | 13.86 | 34.79 | 18.39 | 33.37 | 25.46 |
| LLaMA | 0.33 | 9.15 | 5.41 | 28.10 | 12.26 | 27.72 | 14.70 |
| Claude-3 | 6.12 | 15.86 | 8.15 | 26.56 | 12.17 | 25.50 | 27.57 |
| GPT-4o | 7.15 | 13.28 | 8.41 | 29.50 | 14.13 | 28.43 | 28.18 |
| o4-mini | 3.21 | 19.77 | 11.63 | 28.81 | 14.56 | 27.47 | 25.73 |
| Grok-3 | 7.96 | 28.81 | 18.39 | 37.17 | 20.91 | 35.95 | 36.81 |
| **QwenAB** | **32.13** | **56.23** | **46.50** | **62.57** | **48.13** | **61.81** | **63.55** |
| **LLaMAB** | 30.64 | 54.22 | 43.70 | 62.40 | 47.25 | 61.46 | 63.48 |
| *Mechanism* | | | | | | | |
| Qwen2 | 0.00 | 16.91 | 8.07 | 32.62 | 15.02 | 30.23 | 27.51 |
| LLaMA | 0.00 | 17.89 | 8.52 | 32.84 | 14.67 | 30.84 | 24.72 |
| Claude-3 | 0.07 | 6.84 | 3.34 | 15.76 | 5.95 | 14.68 | 14.95 |
| GPT-4o | 0.33 | 15.05 | 7.68 | 27.94 | 12.39 | 26.59 | 25.49 |
| o4-mini | 0.07 | 8.35 | 3.22 | 18.27 | 6.27 | 16.64 | 15.12 |
| Grok-3 | 0.86 | 26.14 | 13.80 | 37.14 | 18.12 | 34.90 | 35.79 |
| **QwenAB** | 2.56 | 31.06 | 18.51 | 42.50 | 23.63 | 40.30 | 41.52 |
| **LLaMAB** | **2.83** | **33.33** | **19.68** | **45.02** | **25.30** | **42.62** | **44.26** |
| *Function* | | | | | | | |
| Qwen2 | 0.00 | 5.08 | 1.83 | 15.10 | 4.07 | 13.95 | 9.73 |
| LLaMA | 0.00 | 2.93 | 0.95 | 11.76 | 2.56 | 10.95 | 6.16 |
| Claude-3 | 0.00 | 5.16 | 1.82 | 13.64 | 4.01 | 12.48 | 12.41 |
| GPT-4o | 0.37 | 12.03 | 5.11 | 25.56 | 10.40 | 24.16 | 22.69 |
| o4-mini | 0.10 | 7.54 | 2.55 | 16.87 | 4.89 | 15.06 | 14.58 |
| Grok-3 | 0.49 | 16.35 | 6.87 | 29.67 | 12.58 | 28.16 | 26.79 |
| **QwenAB** | **0.88** | **22.65** | **10.74** | **34.52** | **16.37** | **32.41** | **33.75** |
| **LLaMAB** | 0.74 | 21.56 | 10.13 | 33.86 | 15.57 | 31.62 | 33.18 |

## 5.2 FUNCTION GUIDE ANTIBODY DESIGN

The generation of antibodies based on human instructions is a thrilling frontier in scientific research. By integrating antibody design into the capabilities of LLMs, these models can now generate antibody sequences with impressive precision. To evaluate the quality of antibody sequences generated by our fine-tuned models, we adopt two categories of baselines: antibody-specific models, including SeqDesign (Shin et al., 2021), an autoregressive generative model for protein sequence design; IgLM (Shuai et al., 2023), a language model for synthetic antibody library generation; and PALM-

H3 (He et al., 2024), a pre-trained foundation model tailored for antibody sequence design. We also include general LLMs, namely Qwen (Yang et al., 2024), LLaMA (Dubey et al., 2024), GPT-4o (OpenAI et al., 2024), and Claude-3 (Anthropic, 2024), which allow us to assess the ability of LLMs without antibody-specific pretraining to generate plausible antibody sequences. Results for CDR-H3 design are presented in Table 3 and Figure 4, with additional results on antibody *de novo* design provided in Appendix G.2.

We employ the advanced antibody–antigen complex structure prediction method, tFold (Wu et al., 2024), to generate complexes between the modeled antibodies and the target antigen, following the benchmark protocol. As detailed in the evaluation metrics of Appendix F and shown in Table 3, AFD-Instruction consistently outperforms baseline methods in terms of perplexity and SRR, indicating our fine-tuned models generate antibody sequences with greater plausibility and fidelity to natural counterparts. Furthermore, tFold evaluation reveals that AFD-Instruction–tuned models achieve higher pTM, ipTM, and pLDDT scores compared to baseline methods, suggesting enhanced stability, improved inter-chain packing accuracy, and increased confidence in predicted 3D conformations of designed antibody structures.

Table 3: Comparison of models on sequence design and structure metrics. Top-performing results are highlighted in **bold**, while second-best performances are underlined.

| MODEL | PERPLEXITY↓ | SRR↑ | IPTM↑ | PTM↑ | PLDDT↑ |
|---|---|---|---|---|---|
| *Antibody-specific models* | | | | | |
| SeqDesign | 3.021 | 0.823 | 0.408 | 0.647 | 0.784 |
| IgLM | 2.910 | 0.847 | 0.423 | 0.720 | 0.876 |
| PALM-H3 | **2.653** | 0.852 | 0.412 | 0.725 | 0.879 |
| *General LLMs* | | | | | |
| Qwen | 7.595 | 0.163 | 0.439 | 0.691 | 0.806 |
| LLaMA | 10.261 | 0.147 | 0.395 | 0.661 | 0.776 |
| GPT-4o | 3.211 | 0.181 | 0.417 | 0.718 | 0.871 |
| Claude-3 | 3.059 | 0.240 | 0.427 | 0.729 | 0.878 |
| *Ours* | | | | | |
| **QwenAB** | 2.958 | 0.875 | 0.483 | 0.752 | 0.881 |
| **LLaMAB** | 2.857 | **0.884** | 0.476 | 0.744 | 0.880 |
| **GemmaAB** | 2.932 | 0.872 | 0.468 | 0.739 | 0.879 |
| **DeepSeekAB-MoE** | 2.901 | 0.878 | 0.472 | 0.747 | 0.880 |
| **LLaMAB-70B** | 2.871 | 0.883 | **0.491** | **0.759** | **0.883** |

Following the generation of antibody–antigen complexes, we use Rosetta (Koehler Leman et al., 2020) to perform a rapid relaxation procedure that optimizes side-chain conformations and reduces steric clashes, after which the binding free energy ($\Delta G$) is calculated. As shown in Figure 4, lower $\Delta G$ values correspond to more favorable interactions, and the comparison indicates that LLaMAB achieves substantially lower binding $\Delta G$ than IgLM, demonstrating that antibodies generated with AFD-Instruction tuning form stronger and more stable complexes with their target antigens.

## 6 CONCLUSION

In this work, we introduce **AFD-Instruction**, the first large-scale instruction dataset with functional annotations tailored to antibodies, bridging the gap between raw sequence corpora and function-aware supervision. By aligning antibody sequences with concise functional descriptors, AFD-Instruction enables LLMs to infer antibody properties and generate designs under explicit functional guidance. Instruction-tuned models based on this resource achieve substantial improvements in understanding and design tasks, highlighting the importance of incorporating antibody-specific functional signals into LLM training. Moving forward, we aim to expand task coverage, enhance antigen diversity, and implement stricter evaluation protocols that incorporate structural and functional benchmarks, ultimately advancing antibody-focused foundation models for therapeutic discovery.

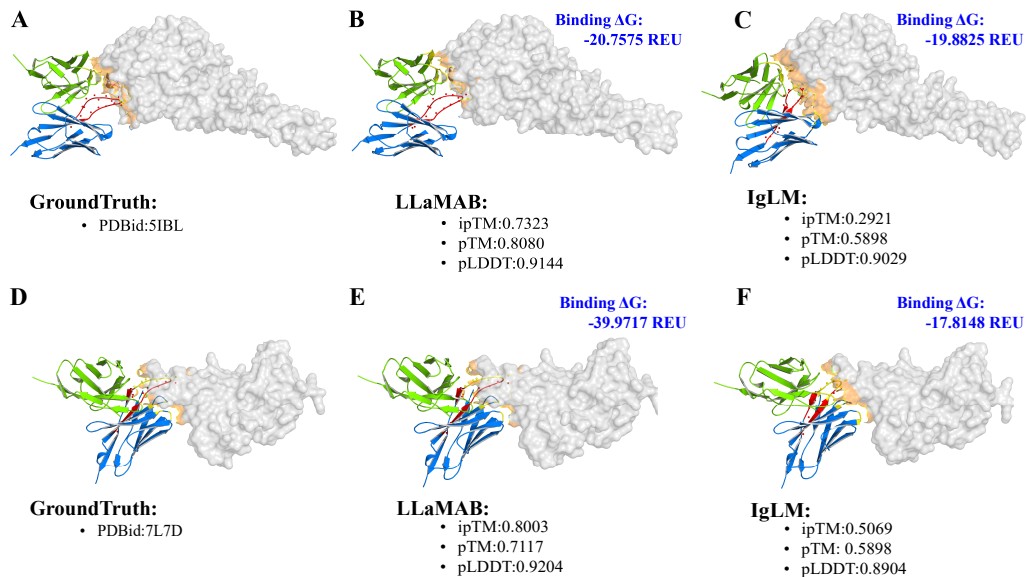

Figure 4: Comparison between LLaMAB and the conventional IgLM framework for antibody sequence design, highlighting the role of functional instruction-based guidance. More visualization results are provided in Appendix Figure 14 and Figure 15.

## ACKNOWLEDGEMENTS

This work is supported by State Key Laboratory of Vaccines for Infectious Diseases, Xiang An Biomedicine Laboratory, China (2025XAKJ0200001), the National Natural Science Foundation of China (32401237), Fujian Provincial Natural Science Foundation of China (2024J08358), the Natural Science Foundation of Xiamen, China (3502Z202371039), Scientific Research Foundation of State Key Laboratory of Vaccines for Infectious Diseases (2024SKLVDzy06).

## ETHICS STATEMENT

All antibody sequences and associated annotations in AFD-Instruction were derived from publicly available sources, including SabDab, PDB, and peer-reviewed scientific literature. No human subject data or personally identifiable information is included. To mitigate intellectual property and licensing concerns, we meticulously ensured that all resources utilized are publicly accessible for research purposes.

Given that AFD-Instruction enables not only antibody understanding but also function-conditioned antibody design, we recognize the potential dual-use risks associated with therapeutic antibody generation. To address these concerns, we implement a controlled release strategy. Specifically, the understanding subset of AFD-Instruction will be made publicly available for non-commercial research under a permissive license, while subsets related to design will be distributed solely upon request and subject to institutional vetting and usage agreements. Furthermore, any pretrained models capable of function-guided sequence generation will be provided exclusively through controlled APIs rather than through bulk parameter releases, enabling us to enforce monitoring and rate-limiting measures.

We further commit to conducting red-teaming and automatic screening processes to identify and mitigate potential misuse scenarios, such as the generation of antibodies targeting high-risk or prohibited pathogens. We urge the community to regard AFD-Instruction strictly as a scientific and educational resource, designed to advance methodologies for safe therapeutic discovery, rather than for unregulated or unsafe experimentation.

## REPRODUCIBILITY STATEMENT

We provide comprehensive descriptions of dataset construction in Appendix C, the training setup in Appendix E, and evaluation metrics in Appendix F. This includes details on the sources of antibody sequences, the processing pipeline, LoRA hyperparameters, hardware configuration, and the definition of all reported metrics. To ensure transparency, we also present extended results and model-specific formatting details in Appendix G.1 and Appendix G.2, enabling researchers to replicate our baseline comparisons. The source code and a partial release of the dataset are available at https://afd-instruction.github.io/. These resources facilitate verification and extension of our work by other researchers.

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

## A LLM USAGE STATEMENT

In accordance with the conference policy on large language models (LLMs), we disclose the use of GPT-5 during manuscript preparation. The model was employed solely for language refinement, including grammatical correction and stylistic polishing of author-written text. All research ideas, experimental design, data analysis, and substantive scientific contributions were conceived and executed by the authors. The LLM did not generate any novel content, hypotheses, or results. Its role was limited to assisting with clarity and readability of the final manuscript.

## B MORE DETAIL OF DATA CONSTRUCTION

Figure 5 presents a representative case of employing the SELF-QUESTIONING strategy to extract mechanism-oriented supervision from literature-derived antibody descriptions using GPT-4o (OpenAI et al., 2024). In this context, the model is explicitly instructed to generate exactly five concise and factual question–answer pairs that remain grounded in the source text, focus exclusively on mechanistic aspects, and avoid direct reference to the antibody name. The resulting pairs highlight essential features such as epitope targeting, the contributions of heavy and light chains, and the functional impact of specific mutations. This example illustrates how self-questioning provides a systematic approach to distilling mechanistic insights into a standardized instructional format, thereby enhancing the construction of reliable supervision signals for antibody understanding tasks.

---

**Prompt:**

You are a precise scientific assistant. From the given paragraph, produce exactly five Question–Answer pairs that are factual, concise, and directly supported by the paragraph. All questions must be about this antibody only, focusing on its mechanism and related aspects, without mentioning the antibody name explicitly. Format the output strictly so that each pair is written as "Q: ... A: ..." with five pairs in total, separated by blank lines, and do not include numbering or any extra text.

Paragraph: The 10E8 mutant 3 antibody functions as a neutralizing monoclonal antibody targeting the MPER epitope of HIV-1 gp41, primarily engaging the epitope through its heavy chain while its light chain interacts with viral membrane lipids. Mutations introducing negative charges on the light chain surface disrupt lipid interactions, significantly reducing neutralization potency and breadth to 34%, underscoring the light chain's critical role in viral neutralization. This antibody highlights the importance of membrane lipid engagement for effective HIV-1 neutralization, though its engineered mutations compromise its functional efficacy. The findings emphasize the delicate balance between epitope binding and membrane interactions in designing antibodies against HIV-1.

---

**GPT-4o completion:**

Q1: What epitope does this antibody target?
A1: It targets the MPER epitope of HIV-1 gp41.

Q2: How does the heavy chain of this antibody contribute to its function?
A2: The heavy chain primarily engages the epitope.

Q3: What role does the light chain play in viral neutralization?
A3: The light chain interacts with viral membrane lipids, which is critical for neutralization.

Q4: How do mutations introducing negative charges on the light chain affect the antibody?
A4: They disrupt lipid interactions, reducing neutralization potency and breadth to 34%.

Q5: What does this antibody's function reveal about HIV-1 neutralization?
A5: It highlights the importance of membrane lipid engagement for effective neutralization.

---

Figure 5: An illustration of SELF-QUESTIONING method for generating factual question–answer pairs based on a given paragraph extracted from the literature in the biomolecular field; the example completion is produced by GPT-4o.

Table 4: Templates for converting structured annotations into textual instructions for CDRH3 design.

| Feature | Template |
|---|---|
| **Strict constraints (goal + mechanism + epitope/affinity/developability)** | |
| Mechanism + Epitope | Create a CDRH3 loop for `{goal}`, acting via `{mechanism}` at `{epitope_or_site}`. |
| Affinity | Design a CDRH3 for `{goal}`, with affinity preference `{affinity_hint}`. |
| Developability | Design a CDRH3 optimized for developability `{developability}`. |
| Specificity | Generate a CDRH3 ensuring specificity `{safety_or_specificity}`. |
| **Moderate constraints (goal + one key attribute)** | |
| Mechanism | Generate a CDRH3 sequence with mechanism `{mechanism}`. |
| Epitope | Propose a CDRH3 targeting epitope `{epitope_or_site}`. |
| Binding site | A CDRH3 that binds ligand at `{binding_site}`. |
| Active site | Complete the CDRH3 ensuring conserved active site `{active_residues}`. |
| Structural bias | A CDRH3 biased toward `{structural_motif}`. |
| Topology | CDRH3 topology set to `{topology}` (e.g., loop length, flexibility). |
| Safety | CDRH3 designed to avoid cross-reactivity with `{off_targets}`. |
| **Loose constraints (goal only)** | |
| Goal | Design a CDRH3 to achieve `{goal}`. |

In addition to self-questioning, we employ a *template-based conversion* strategy. This approach involves the direct transformation of structured metadata extracted from publications or databases into instruction–response pairs through predefined templates, thereby ensuring the faithful preservation of factual content. Specifically, Table 4 presents a collection of templates tailored for CDRH3 design, illustrating how diverse functional attributes (e.g., mechanism, epitope, binding site, developability) can be systematically mapped into natural language instructions. To maintain biological plausibility, any generations that deviate from the intended functionality or exhibit structural inconsistencies are filtered out. Consequently, this design-oriented dataset complements the understanding tasks, enabling large language models not only to analyze antibody properties but also to conditionally generate novel candidates guided by explicit functional intent.

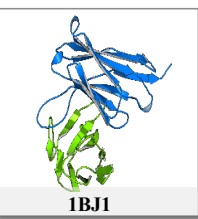

**1BJ1**

The antibody functions as a neutralizing monoclonal antibody by sterically blocking VEGF interactions with its receptors KDR and Flt-1, specifically targeting outer strands β5 and β6 near the β turn without inducing conformational changes in VEGF. This mechanism effectively inhibits pathogenic angiogenesis, making it a promising therapeutic for cancer and diabetic retinopathy. Notably, the antibody achieves high-affinity binding through a functionally complementary interface, with its heavy chain dominating the buried surface area, while selectively stabilizing a specific VEGF loop conformation. In vivo evidence supports its efficacy in suppressing tumor growth, underscoring its potential as a targeted anti-angiogenic treatment.

This antibody variant achieves enhanced affinity for turkey egg-white lysozyme (TEL) through subtle global structural adjustments rather than localized complementarity improvements at the mutation site. These adjustments increase the VH–VL domain interaction surface, thereby stabilizing the antigen–antibody complex. The binding epitope involves key TEL residues Trp62, Lys73, and Asn77, and thermodynamic analyses indicate that the affinity enhancement is primarily driven by favorable enthalpic changes. These findings underscore how nonlocal structural reorganization can compensate for mutations, providing insights into antibody engineering and antigen recognition.

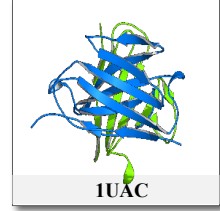

**1UAC**

Figure 6: Examples of natural language descriptions within our antibody dataset, where structural representations are paired with textual explanations of binding mechanisms and functional properties.

Representative cases of error correction are illustrated in Figure 7. These examples highlight the effectiveness of our quality control (QC) rules in addressing issues related to chain attribution, domain mapping, residue-level validation, and consistency within therapeutic scope. Collectively, these measures enhance the automated checks and expert reviews outlined in Section 3.3, demonstrating a systematic approach to identifying and rectifying errors.

| Case (PDB) | Observed Extraction Error) → QC Rule → Corrected |
|---|---|
| **5T29** | *Observed Extraction Error*: "**Light chain** primarily engages the MPER epitope; **heavy chain** plays a minor role. Negative-charge mutations on the **heavy chain** reduce neutralization to 34%." 
 *QC rule*: chain–epitope attribution from structure (heavy vs. light), mutation-site consistency, lipid-interaction evidence. 
 *Corrected*: "**Heavy chain primarily engages the epitope.** The **light chain** interacts with viral membrane lipids, which is critical for neutralization. Mutations introducing negative charges on the **light chain** disrupt lipid interactions, **reducing neutralization potency and breadth to 34%**. This highlights the importance of membrane lipid engagement for effective neutralization." |
| **1S78** | *Observed Extraction Error*: "Targets **EGFR** and binds **domain IV**; effective across all cancers; identical to trastuzumab in mechanism." 
 *QC rule*: domain mapping, claim scope aligned to clinical evidence, therapy differentiation. 
 *Corrected*: "**Targets ErbB2**. It **binds domain II of ErbB2**, sterically blocking the **dimerization interface**. It is **clinically relevant in HER2-positive breast cancer**. Unlike trastuzumab, it **prevents heterodimerization without inhibiting ErbB2 shedding**, thereby **disrupting interactions with partner receptors and inhibiting downstream signaling**." |
| **6HJP** | *Observed Extraction Error*: "Targets the **HA receptor-binding site** and broadly neutralizes **all** influenza subtypes by blocking sialic-acid binding." 
 *QC rule*: epitope location (ectodomain–membrane junction vs. RBS), subtype specificity, mechanism wording grounded in structure. 
 *Corrected*: "**Targets the junction between the ectodomain and membrane anchor of influenza HA**. It **restricts the flexibility of the HA linkage**, interfering with conformational changes needed for membrane fusion. It **sterically constrains the ectodomain orientation relative to the transmembrane domain**, a previously unrecognized mode. It **specifically targets H1 influenza viruses**, underscoring clinical relevance to this pathogen." |
| **3OPZ** | *Observed Extraction Error*: "**Occupies the catalytic pocket** to block enzymatic activity; distal residues are irrelevant." 
 *QC rule*: catalytic vs. distal binding-site validation, residue-level mechanism (Y119 mobility), functional consequence alignment. 
 *Corrected*: "**Restricts the mobility of a critical assisting tyrosine (Y119)**, **disrupting enzymatic function without blocking the catalytic site**. It **inhibits sialic-acid transfer** to parasite surfaces, reducing cellular invasion, and **mitigates immune and hematological dysregulation**. It is **effective against Chagas' disease** and **targets distal non-catalytic residues**, offering a refined inhibition strategy." |
| **5JXE** | *Observed Extraction Error*: "Binds **outside** the ligand-binding site (no overlap), acts mainly via Fc-mediated ADCC, and is effective against **all solid tumors**." 
 *QC rule*: ligand-overlap mapping (C′D loop), primary MoA vs. effector functions, indication scope per evidence. 
 *Corrected*: "**Blocks the interaction between PD-1 and its ligands PD-L1/PD-L2**, restoring T-cell activation and antitumor responses. It **binds the C′D loop of PD-1**, **overlapping with the ligand-binding site**, and **stabilizes** this otherwise disordered loop. Clinically, it is effective against **melanoma, lymphoma, and non-small-cell lung cancer**." |

Figure 7: Representative error types encountered during antibody–function pair construction. For each case, we show an erroneous automatic extract, the quality-control rule used to identify the issue, and the corrected record. These examples illustrate how our QC pipeline enforces factual grounding, chain attribution, and therapeutic scope consistency.

## C   A MORE EXHAUSTIVE DATA ANALYSIS OF AFD-INSTRUCTION

**Global Length and Composition**    We first investigated the overall distribution of CDR-H3 sequences within the AFD-Instruction. Figure 8 (top) shows the length distribution, which is centered around a median of 14 amino acids and exhibits a long tail indicative of extended loops. This obser-

Table 5: Comparison with representative protein–language datasets.

| DATASETS | # PAIRS | USAGE | ACCESS |
|---|---|---|---|
| *General Protein Domain* | | | |
| Swiss-Prot (Consortium, 2023) | ∼573k | Function notes (seq→text) | Open |
| SwissProt-Aug (Yuan et al., 2024) | ∼4M | Description augmentation | Mixed |
| ProteinKG25 (Zhang et al., 2022b) | ∼5.0M links | Ontology-aware pretraining | Open |
| *Enzyme / Structure Focus* | | | |
| Enzyme (BioTalk) (Hoarfrost et al., 2024) | 22k–28M | Enzyme function captioning | Open |
| ProteinChat (Guo et al., 2023) | ∼1.5M | Structure/function summaries | Open |
| *Instruction-Oriented* | | | |
| Evola (Zhou et al., 2025) | ∼546M Q&A | Instruction tuning (protein Q&A) | Closed |

vation consistents with repertoire studies conducted on human and murine antibodies. The global amino acid composition, as depicted in Figure 8 (bottom left), is predominantly characterized by Tyr, Asp, Gly, Ala, Arg, and Ser. This finding corroborates previous analyses that have reported frequent usage of Tyr/Gly/Ser in CDR-H3 repertoires (Zemlin et al., 2003). Collectively, these global statistics affirm that the dataset accurately reflects realistic features at the repertoire level.

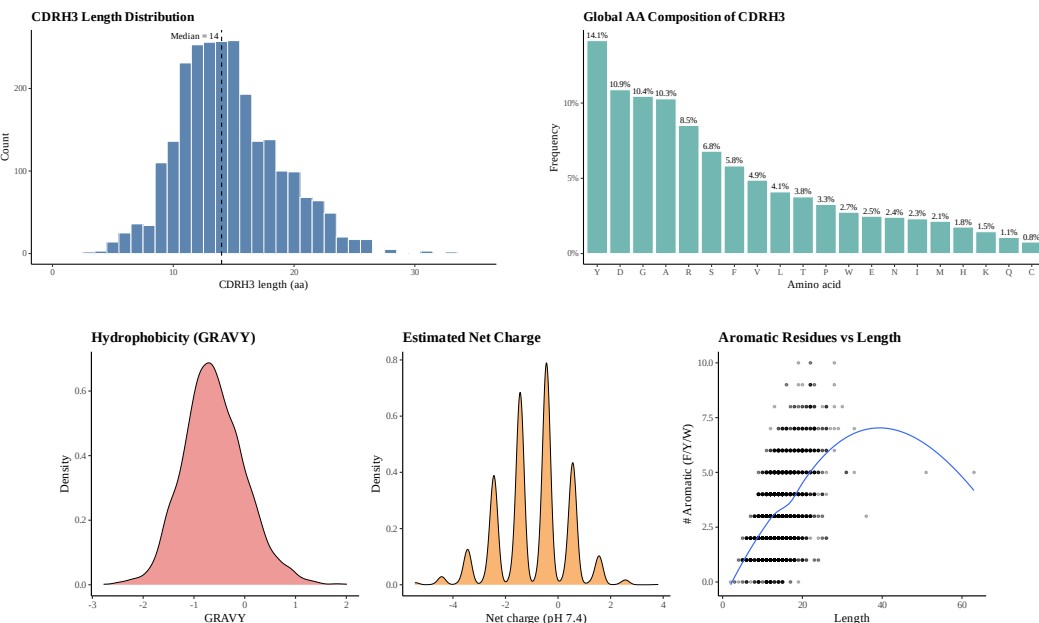

Figure 8: Global characteristics of CDR-H3 sequences in AFD-Instruction are presented, including the length distribution (left), amino acid composition (right), and physicochemical properties (bottom). These analyses demonstrate that the dataset accurately reflects known repertoire-level features.

**Physicochemical Validity** We subsequently evaluated the physicochemical properties both globally and in relation to CDR-H3 length. Figure 8 (bottom right) illustrates the distributions of hydrophobicity and net charge, with values clustering around neutral to moderately hydrophilic and neutral to moderately positive ranges, respectively. This observation aligns with repertoire-level constraints. Figure 9 expands upon this analysis by stratifying sequences according to their lengths. Longer CDR-H3 loops exhibit a higher content of aromatic and moderately basic residues, while shorter loops are enriched in glycine and occasional proline, reflecting their roles in loop stability and flexibility (Shinoda & Fujitani, 2017; Regep et al., 2017). Two-dimensional density plots further demonstrate that extended loops show greater variance in charge and hydrophobicity profiles, exhibiting a trend towards increased hydrophobicity and positive charge. This finding is consistent with structural studies of antibodies targeting membrane-proximal epitopes (Zhang et al., 2019; Irimia et al., 2016; 2017). Together, these analyses indicate that AFD-Instruction effectively reproduces the realistic sequence–property coupling observed in natural repertoires, thereby supporting its application as a foundational tool for understanding antibody characteristics and guiding design tasks.

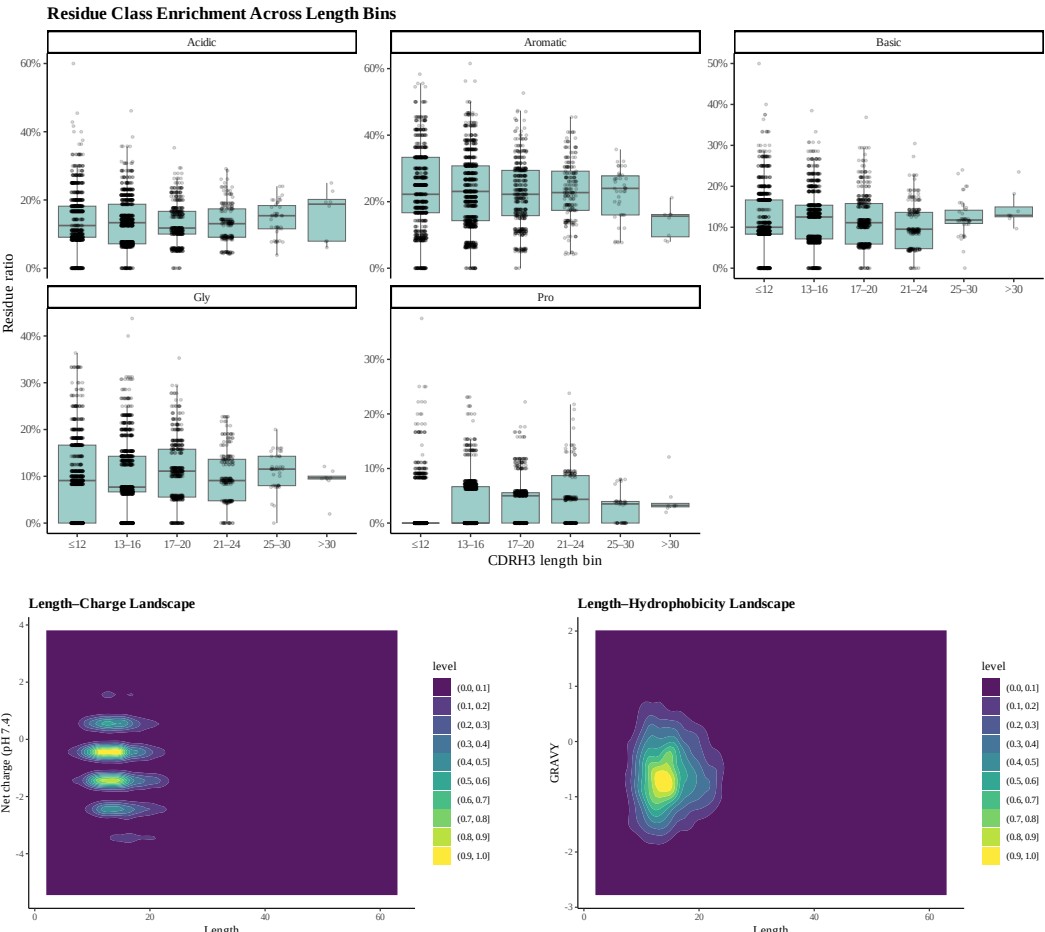

Figure 9: Dataset composition and physicochemical validity of CDR-H3 sequences in AFD-Instruction. Top: Residue-class enrichment across length bins indicates an increased presence of aromatic and basic residues in longer loops, alongside a notable enrichment of glycine and proline in shorter loops. Two-dimensional density plot illustrating the relationship between length and net charge reveals that most sequences cluster around neutral to moderately positive values, with greater variance observed in extended loops. Bottom right: Two-dimensional density plot depicting the correlation between length and hydrophobicity (GRAVY) demonstrates a trend towards more hydrophobic profiles in longer loops. Collectively, these analyses confirm that the dataset reflects biologically plausible sequence–property coupling.

**Description Case**    In addition to providing global statistics, we further illustrate the quality of the dataset through specific descriptive cases. Figure 6 presents examples in which structural representations of antibodies are accompanied by detailed natural language descriptions. These descriptions effectively capture binding mechanisms, critical residues, and functional consequences while ensuring consistency with experimentally validated structural insights. This alignment between structural context and textual annotation underscores the reliability of AFD-Instruction as a valuable resource for training models aimed at understanding antibody function and design.

**Instruction Diversity and Semantic Structure**    To characterize the semantic diversity of AFD-Instruction, we randomly sampled a subset of instructions from the complete dataset. We then embedded their texts using an MPNet-based sentence encoder within the Sentence-BERT framework (Reimers & Gurevych, 2019), subsequently projecting the resulting vectors with UMAP, as illustrated in Figure 10. The embedding space produced is organized into numerous distinct semantic clusters rather than converging into a limited number of highly dense regions, indicating a wide range of instruction intents.

Applying KMeans clustering with thirty clusters results in cluster sizes that range from 170 to 971 instructions, with a median of 562. The entropy of the cluster size distribution is measured at 3.33 nats, which approaches the theoretical maximum of approximately 3.40 nats for thirty equally sized clusters. This suggests that no small subset of clusters dominates the corpus. A nearest-neighbor analysis conducted on a randomly sampled subset of 4,000 instructions from the full dataset reveals a mean cosine similarity of 0.83 and a median of 0.84, with no pairs exceeding a similarity score of 0.95; this finding argues against significant large-scale paraphrastic duplication. Furthermore, instructions associated with the same antibody demonstrate higher semantic similarity compared to those across different antibodies, exhibiting average similarities of 0.595 for intra-PDB comparisons and 0.506 for inter-PDB comparisons respectively. Collectively, these observations indicate that the variability observed in AFD-Instruction is primarily driven by biologically grounded differences among antigens and mechanisms rather than superficial linguistic rephrasing.

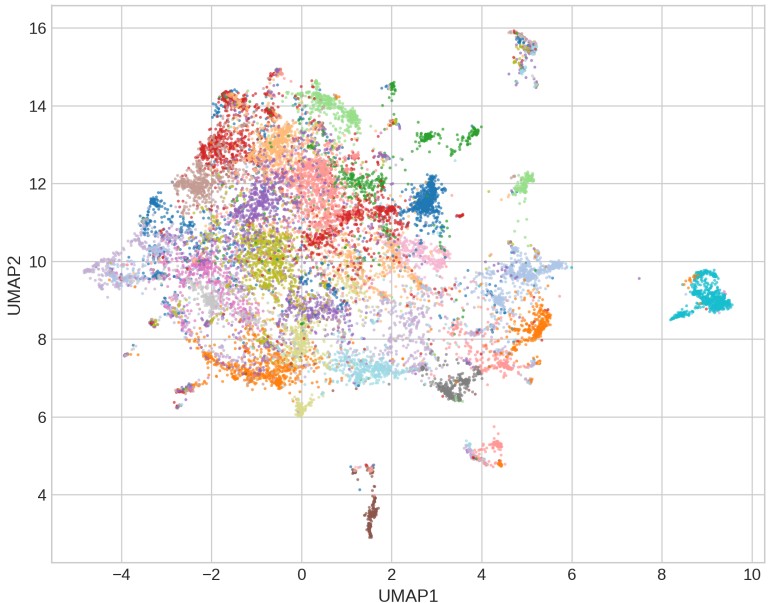

Figure 10: UMAP projection of instruction embeddings derived from the complete instruction text and clustered using $K = 30$ KMeans. The embedding space reveals multiple well-defined semantic regions with balanced coverage, indicating a broad diversity of instructional intents rather than dominance by a limited number of paraphrastic templates.

## D  COMPARISON OF PROTEOME-WIDE AND ANTIBODY-SPECIFIC SUPERVISION

**Proteome-wide instruction datasets.** Evola is designed as a proteome-wide functional Q&A model: its training data consist of large-scale AI-generated protein–question–answer triples derived from UniProt and related resources, targeting generic protein annotation such as EC numbers, Gene Ontology terms, subcellular localization, and disease association. Each UniProt entry is treated as an individual protein, and supervision is provided at the level of high-level functional descriptions associated with that entry. Immunoglobulin entries constitute only a small fraction of this space, and many of them correspond to germline-like V/J/C segments rather than complete variable regions with paired chains.

**Antibody-specific supervision in AFD-Instruction.** In contrast, AFD-Instruction is explicitly antibody-centric. Each instance aligns a mature, fully paired antibody variable region (VH and VL, including complete CDR3) to concise, normalized descriptors of target antigen, disease context, binding site, mechanism, and functional activity, and also includes function-guided design instructions (e.g., generating variable regions or CDR3s conditioned on antigen and desired activity). This

supervision is therefore defined on antibody–antigen–sequence relationships and downstream design behaviour, rather than on per-protein functional labels.

**Biological differences between general proteins and antibodies.**     From a biological standpoint, antibody function and specificity differ qualitatively from typical protein-function labels. Antibodies are modular heterodimers: the same heavy chain can exhibit very different specificity when paired with different light chains, and many UniProt immunoglobulin records describe isolated germline segments rather than mature VH/VL pairs. In particular, the heavy-chain CDR3 is the most diverse and often the most critical determinant of the paratope; small sequence changes in CDR3 can dramatically alter antigen specificity and affinity, and many practical engineering tasks explicitly operate at the CDR3 level. Generic protein annotations such as "antigen binding" or broad immune functions rarely encode explicit antigen sequences, epitope-level binding sites, neutralization or blocking mechanisms, or CDR-level design constraints. As a result, proteome-wide instruction corpora do not provide the paired, antigen-aware, CDR3-resolving supervision needed for antibody-specific interpretation and design.

## E    EXPERIMENTAL SETUP

Based on the AFD-Instruction, we employ Qwen-7B and LLaMA-8B as foundation models throughout our experiments and conduct instruction tuning on them. To address the complexity of the task, we partition the data by subtask and train the models independently on each subset. For evaluation, approximately 2,000 samples are reserved as a test set for each subtask. To ensure uniqueness between train and test sets and avoid information leakage arising from common sequence motifs in antibody repertoires, we first cluster all antibodies using MMseqs2 (Steinegger & Söding, 2017) at an 80% sequence identity threshold and then enforce that sequences from the same cluster are placed entirely within either the training or the test split. This redundancy-aware partitioning ensures that highly similar antibodies do not cross splits, while maintaining balanced distributions of sequence length and amino acid composition across subsets, thereby supporting both fairness and reliability in performance assessment.

We employ Low-Rank Adapter (LoRA) fine-tuning in our experiments, an efficient approach that reduces memory consumption during training by updating only a small subset of parameters while keeping the pre-trained model weights fixed. All experiments are conducted using PyTorch on two H100 GPUs with 80 GB of memory each. We set the batch size to 8 per GPU and adopt a context length of 2,048 tokens. The specific hyperparameters tuned for each model are summarized in Table 6.

Table 6: LoRA Fine-Tuning Experimental Parameters

| Parameter | Value |
|---|---|
| *LoRA Hyperparameters* | |
| LoRA Rank ($r$) | 16 |
| LoRA Alpha | 16 |
| LoRA Dropout | 0.0 |
| Target Modules | q_proj, k_proj, v_proj, up_proj, down_proj, o_proj, gate_proj |
| *Training Hyperparameters* | |
| Learning Rate | $3.0 \times 10^{-4}$ |
| LR Scheduler | Linear |
| Batch Size (per GPU) | 8 |
| Gradient Accumulation | 2 |
| Epochs | 1 |
| Mixed Precision | BF16 |
| Optimizer | AdamW |
| Weight Decay | 0.01 |
| Warmup Steps | 10 |

# F    EVALUATION METRICS

**Antibody Sequence Understanding**    In the context of binary classification tasks, we utilize accuracy (Acc) and F1 score as our primary evaluation metrics. Accuracy provides a measure of the overall correctness of predictions, whereas the F1 score offers a balanced assessment of precision and recall, making it a more robust metric in scenarios characterized by class imbalance. Consistent with ProtT3 (Liu et al., 2024b), we evaluate antibody captioning performance through several metrics including exact match, BLEU (Papineni et al., 2002), ROUGE (Lin, 2004), and METEOR (Banerjee & Lavie, 2005). These metrics assess the quality of generated outputs by comparing them against reference answers.

**Function-Guided Antibody Design**    To evaluate the rationality of the generated antibody sequences, we introduce Sequence Recovery Rate (SRR) (Shin et al., 2021) as a metric to assess both the diversity of the generated sequences and their similarity to natural antibody sequences. Additionally, we compute Perplexity using IgLM (Shuai et al., 2023) to evaluate how well the generated antibody sequences conform to the learned distribution of natural antibodies, thereby reflecting their fluency and plausibility. After predicting the antibody–antigen complex structures with tFold (Wu et al., 2024), we utilize the method to evaluate the predicted template modeling score (pTM), the interface pTM (ipTM) (Evans et al., 2022), and the predicted local distance difference test (pLDDT) (Jumper et al., 2021) across different approaches. The pTM and ipTM estimate the likelihood that the modeled antibody binds to the correct epitope and forms a stable complex, whereas the pLDDT provides a confidence measure for the predicted antibody structure.

# G    ADDITIONAL RESULTS

Given the page limit, only representative results are shown in the main text, while additional results demonstrating the effectiveness of our AFD-Instruction are provided in Figure 11, Figure 12, Figure 14, Figure 15, Figure 16, Figure 17, Figure 18, and Table 7. Note that ProtT3 (Liu et al., 2024b) and ProtLLM (Zhuo et al., 2024) require task-specific input formats, whereas BioT5 (Pei et al., 2023) achieves its anticipated performance solely when sequences are explicitly enclosed by <bop> and <eop> tokens. To facilitate a fair evaluation, we converted our instruction formats into the model-specific formats required. Additionally, it is noteworthy that ProtLLM is primarily designed for classification-style tasks; therefore, we did not include it as a baseline for captioning tasks.

## G.1    CAPTION RESULT

Table 7 expands upon the results of the captioning task by incorporating *Domain-Specific Protein Models* and additional open-source baselines. Overall, domain-specific models exhibit challenges in generalizing beyond narrow sequence-level objectives, whereas general-purpose large language models (LLMs) fine-tuned with **AFD-Instruction** consistently demonstrate superior performance across all categories. In addition to quantitative scores, Figure 17 further elucidates this contrast: protein-focused models frequently yield fragmented or irrelevant outputs, while instruction-tuned LLMs such as QwenAB and LLaMAB produce accurate and interpretable descriptions that closely align with ground truth. These descriptions effectively cover binding sites, immune responses, and mechanistic motifs. These case studies complement the benchmark results and illustrate how AFD-Instruction facilitates a connection between protein-level modeling and antibody-specific functional reasoning.

## G.2    ANTIBODY DESIGN RESULT

The distributions of *de novo* designed antibody sequences generated by AFD-Instruction fine-tuned LLaMAB and QwenAB are presented in Figure 11, encompassing both heavy (left) and light (right) chains. Compared to the ground truth, both models produce sequence distributions that substantially overlap with those of natural antibodies, demonstrating their ability to capture the global structural characteristics of antibody sequence space. Notably, QwenAB exhibits broader coverage of the ground truth distribution, whereas LLaMAB produces more compact clusters, suggesting distinct generative biases between the two models. These findings underscore that AFD-Instruction enables

Table 7: Extended caption task results for antibody sequence understanding. This table reports the performance of *Domain-Specific Models* together with the remaining *Open-Source General-Purpose LLMs*, following the same evaluation protocol as Table 2.

| MODEL | EXACT↑ | BLEU-2↑ | BLEU-4↑ | ROUGE-1↑ | ROUGE-2↑ | ROUGE-L↑ | METEOR↑ |
|---|---|---|---|---|---|---|---|
| *Binding* | | | | | | | |
| Galactica | 0.00 | 3.35 | 0.99 | 14.39 | 3.56 | 13.67 | 10.27 |
| Mol-Instructions | 0.00 | 2.61 | 0.68 | 13.29 | 3.10 | 11.13 | 17.74 |
| BioT5 | 0.00 | 0.44 | 0.04 | 0.94 | 0.01 | 0.94 | 2.44 |
| InstructProtein | 0.00 | 0.03 | 0.01 | 0.74 | 0.01 | 0.72 | 0.20 |
| ProtT3 | 0.00 | 0.02 | 0.01 | 1.56 | 0.02 | 1.56 | 0.53 |
| OPT | 0.00 | 9.31 | 3.30 | 20.42 | 4.67 | 19.34 | 13.07 |
| Alpaca | 0.00 | 13.46 | 5.71 | 23.98 | 7.62 | 22.57 | 18.51 |
| gemma-3 | 0.00 | 4.63 | 1.54 | 17.51 | 2.97 | 16.98 | 9.16 |
| DeepSeek-V3 | 1.35 | 17.50 | 7.17 | 25.64 | 7.93 | 24.56 | 26.54 |
| GemmaAB | 5.64 | 32.09 | 19.72 | 41.74 | 22.36 | 40.40 | 41.44 |
| DeepSeekAB-MoE | 5.71 | 36.74 | 24.25 | 48.55 | 29.71 | 46.26 | 47.55 |
| LLaMAB-70B | 5.78 | 33.24 | 19.89 | 43.50 | 23.13 | 42.21 | 42.83 |
| *Disease* | | | | | | | |
| Galactica | 0.09 | 3.78 | 1.71 | 15.58 | 6.20 | 15.13 | 10.31 |
| Mol-Instructions | 0.00 | 1.80 | 0.49 | 10.79 | 2.35 | 9.33 | 16.25 |
| BioT5 | 0.00 | 0.15 | 0.02 | 1.07 | 0.00 | 1.06 | 2.31 |
| InstructProtein | 0.00 | 0.02 | 0.00 | 0.68 | 0.00 | 0.67 | 0.16 |
| ProtT3 | 0.00 | 0.08 | 0.02 | 2.23 | 0.13 | 2.23 | 0.79 |
| OPT | 0.21 | 9.50 | 5.44 | 23.96 | 8.60 | 23.32 | 12.61 |
| Alpaca | 0.33 | 16.37 | 7.59 | 29.29 | 11.92 | 27.59 | 21.67 |
| gemma-3 | 0.30 | 3.38 | 2.38 | 22.36 | 9.88 | 21.87 | 11.18 |
| DeepSeek-V3 | 8.66 | 19.74 | 11.81 | 33.18 | 16.73 | 32.36 | 32.32 |
| GemmaAB | 35.66 | 59.01 | 49.26 | 66.48 | 52.06 | 65.60 | 66.95 |
| DeepSeekAB-MoE | 34.17 | 57.58 | 47.91 | 64.26 | 50.08 | 63.43 | 64.99 |
| LLaMAB-70B | 34.74 | 60.01 | 49.24 | 68.24 | 52.50 | 67.26 | 68.70 |
| *Mechanism* | | | | | | | |
| Galactica | 0.00 | 5.96 | 2.11 | 19.68 | 6.32 | 18.34 | 14.82 |
| Mol-Instructions | 0.00 | 2.61 | 0.68 | 13.29 | 3.10 | 11.13 | 17.74 |
| BioT5 | 0.00 | 0.19 | 0.01 | 0.84 | 0.01 | 0.83 | 1.68 |
| InstructProtein | 0.00 | 0.02 | 0.00 | 1.14 | 0.01 | 1.12 | 0.16 |
| ProtT3 | 0.00 | 0.00 | 0.00 | 0.75 | 0.01 | 0.75 | 0.26 |
| OPT | 0.00 | 10.35 | 4.34 | 23.54 | 6.82 | 22.16 | 15.03 |
| Alpaca | 0.00 | 15.26 | 6.71 | 30.46 | 12.62 | 28.02 | 25.34 |
| gemma-3 | 0.00 | 6.41 | 3.03 | 25.04 | 8.10 | 24.06 | 14.39 |
| DeepSeek-V3 | 0.42 | 17.51 | 8.21 | 29.87 | 11.51 | 28.06 | 27.15 |
| GemmaAB | 4.55 | 35.68 | 22.71 | 48.03 | 28.88 | 45.80 | 46.66 |
| DeepSeekAB-MoE | 4.11 | 34.71 | 21.82 | 46.85 | 28.00 | 44.69 | 45.75 |
| LLaMAB-70B | 5.92 | 38.58 | 25.46 | 50.98 | 31.20 | 48.57 | 49.93 |
| *Function* | | | | | | | |
| Galactica | 0.00 | 2.92 | 0.67 | 9.83 | 1.66 | 9.01 | 6.25 |
| Mol-Instructions | 0.00 | 1.61 | 0.27 | 10.19 | 1.48 | 8.15 | 13.98 |
| BioT5 | 0.00 | 0.17 | 0.02 | 0.96 | 0.01 | 0.94 | 1.68 |
| InstructProtein | 0.00 | 0.02 | 0.00 | 1.37 | 0.00 | 1.35 | 0.25 |
| ProtT3 | 0.00 | 0.00 | 0.00 | 0.88 | 0.03 | 0.88 | 0.27 |
| OPT | 0.00 | 2.45 | 0.67 | 11.51 | 2.04 | 10.85 | 5.86 |
| Alpaca | 0.00 | 8.70 | 2.64 | 19.57 | 4.99 | 17.86 | 13.82 |
| gemma-3 | 0.00 | 0.74 | 0.24 | 10.18 | 2.34 | 9.70 | 4.69 |
| DeepSeek-V3 | 0.33 | 13.86 | 5.68 | 27.52 | 10.78 | 26.06 | 24.57 |
| GemmaAB | 1.21 | 24.26 | 12.39 | 36.88 | 18.24 | 34.49 | 35.35 |
| DeepSeekAB-MoE | 1.49 | 23.46 | 11.82 | 35.7 | 17.69 | 33.6 | 34.39 |
| LLaMAB-70B | 1.46 | 24.03 | 11.93 | 37.06 | 17.84 | 34.82 | 35.90 |

LLMs to generate antibody sequences that are not only diverse but also consistent with the distributions observed in naturally occurring repertoires.

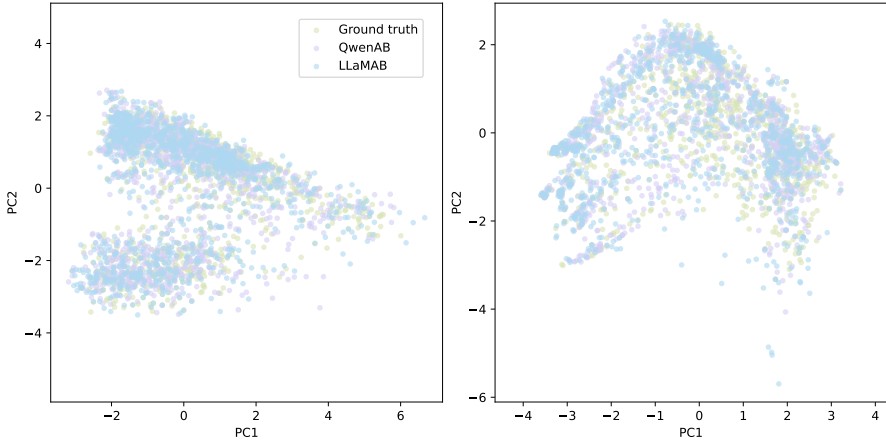

Figure 11: Distribution of antibody sequences for heavy (left) and light (right) chains. We obtained latent representations of each antibody sequence using IgLM (Shuai et al., 2023) and then applied PCA (Shlens, 2014) for dimensionality reduction, allowing us to discern the distributional differences between generated antibody sequences and the ground truth.

Beyond global distributional patterns, we further examine residue-level features of the generated antibodies by comparing sequence logos of the CDR-H3 regions between LLaMAB-generated and natural antibodies (Figure 12). Both artificial and natural repertoires share conserved motifs at the N-terminal positions, most prominently the strong enrichment of A and R, which reflects canonical germline-encoded patterns. At the C-terminal end, both repertoires display a characteristic F-D-Y motif (hydrophobic–acidic–aromatic) with comparable prominence rather than a single-residue dominance. This indicates that LLaMAB effectively captures positional preferences and conserved motifs of CDR-H3, including the canonical F-D-Y pattern. Collectively, these observations demonstrate that AFD-Instruction enables models not only to approximate global antibody sequence distributions but also to reproduce residue-level conservation patterns characteristic of natural repertoires.

As shown in Table 3, general LLMs such as GPT-4o and Claude-3 achieve relatively high ipTM, pTM, and pLDDT scores, suggesting their capability to generate structurally stable antibody conformations. However, their SRR values remain substantially lower than those of antibody-specific or AFD-Instruction–tuned models, indicating that their designed CDR-H3 sequences deviate from natural antibody repertoires. Consistently, Figures 14 and 15 reveal that baseline LLM designs generally exhibit higher binding free energies ($\Delta G$), which reflect weaker and less stable antigen interactions. In brief, these results imply that while general LLMs can produce folded structures with reasonable stability, they lack the repertoire consistency and binding specificity captured by antibody-focused models.

In addition to CDR-H3 design, we further conducted full *de novo* antibody generation with our fine-tuned models. As shown in Figure 16, both LLaMAB and QwenAB produce well-folded complexes with consistently high pLDDT scores, confirming reliable structural stability. Importantly, LLaMAB consistently achieves lower binding free energies ($\Delta G$), indicating stronger and more stable antigen interactions, whereas QwenAB also yields structurally plausible antibodies but with comparatively weaker binding affinities. These results demonstrate that AFD-Instruction enables our fine-tuned models to generate antibodies that are not only structurally sound but also exhibit favorable binding properties in the *de novo* setting.

### G.3 EVALUATION ON A TEMPORAL HOLD-OUT ANTIBODY TEST SET

A concern with the high performance of specialized models on the original benchmark is their potential reliance on spurious correlations or template-like regularities instead of robust biological understanding. To mitigate this confounding factor and more rigorously assess temporal generalization, we constructed a temporal out-of-distribution (OOD) test set from antibodies newly released in the PDB after September 2025. Specifically, we collected 33 antibody entries listed in Table 8, all of

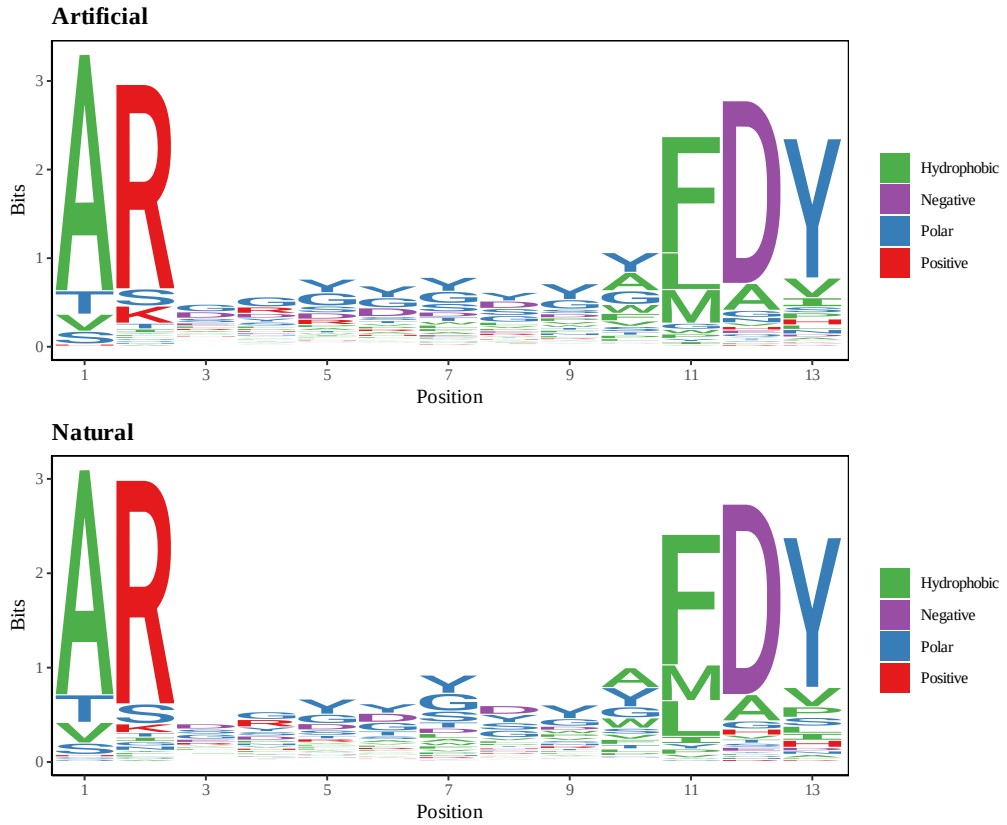

Figure 12: Sequence logos of CDR-H3 regions comparing LLaMAB generated (Artificial) and ground truth (Natural) antibody sequences. These logos illustrate the positional preferences of amino acids, emphasizing conserved motifs and highlighting distributional differences between generated and natural repertoires.

which are supported by primary literature. For each PDB entry, we examined the associated publication and curated question–answer pairs covering the same five dimensions of understanding as those used in the main benchmark: binding site, class, disease, function, and mechanism. All models were subsequently evaluated on this new set using exactly the same prompts and decoding configuration as employed in the original experiments. Given that these complexes and their annotations post-date both the construction of the benchmark and the development of QwenAB and LLaMAB, this temporal split mitigates potential influences from memorization or template matching while providing a more stringent assessment of generalization to newly emerging antibodies. The results obtained from this temporal OOD benchmark are summarized in Table 9.

## G.4 CASE STUDY

To further examine whether the model responds meaningfully to functional natural language instructions, we design a set of controlled case studies that isolate the influence of the prompt while holding all other generation conditions fixed. Each case introduces two contrasting functional directives that reflect well established design considerations in antibody engineering, such as modulating biophysical properties of the CDR loops or directing binding toward distinct antigenic regions. For every pair of directives, the model is prompted with full natural language descriptions and asked to generate antibody variants under identical sampling settings. The resulting sequences are then subjected to standardized biophysical and structural analyses in order to assess whether systematic differences emerge that correspond to the intended functional semantics. This setup enables a direct evaluation of instruction sensitivity independent of model stochasticity or prior sequence biases.

Table 8: Antibody structures that are deposited in the Protein Data Bank after September 2025 serve as the foundation for the temporal out-of-distribution test set. This test set is utilized to evaluate the generalization capabilities of models when applied to newly emerging antibody complexes.

| PDB ID | Release date | PDB ID | Release date |
|---|---|---|---|
| 9dd6 (Roark et al., 2025) | 2025-09-10 | 9mqr (Hattori et al., 2025) | 2025-09-24 |
| 9vhz (Zhao et al., 2025) | 2025-10-01 | 9ei5 (Davoudinasab et al., 2025) | 2025-09-03 |
| 9kur (Xue et al., 2025) | 2025-10-15 | 9n5i (Bekkering et al., 2025) | 2025-10-15 |
| 9ubq (Tao et al., 2025) | 2025-10-29 | 9jzq (Wang et al., 2025) | 2025-09-03 |
| 9dbx (Ouyang et al., 2025) | 2025-09-03 | 9pkc (Khan et al., 2025) | 2025-10-01 |
| 9lf8 (Zhao et al., 2025) | 2025-10-01 | 9pkf (Khan et al., 2025) | 2025-10-01 |
| 9mso (Fantin et al., 2025) | 2025-09-10 | 9n5h (Bekkering et al., 2025) | 2025-10-15 |
| 9lrf (Hiura et al., 2025) | 2025-10-15 | 9sg0 (Ungan et al., 2025) | 2025-10-29 |
| 9sg2 (Ungan et al., 2025) | 2025-10-29 | 8yy6 (Sun et al., 2025) | 2025-10-15 |
| 9msp (Fantin et al., 2025) | 2025-09-10 | 9lra (Hiura et al., 2025) | 2025-10-15 |
| 9sgh (Ungan et al., 2025) | 2025-10-29 | 9pkv (Khan et al., 2025) | 2025-10-01 |
| 9px5 (da Silva Almeida et al., 2025) | 2025-09-10 | 9k3t (Harada et al., 2025) | 2025-10-15 |
| 9sfx (Ungan et al., 2025) | 2025-10-29 | 9pix (Khan et al., 2025) | 2025-10-01 |
| 9n5o (Bekkering et al., 2025) | 2025-10-15 | 9msn (Fantin et al., 2025) | 2025-09-10 |
| 9s8w (Barash et al., 2025) | 2025-09-24 | 9p97 (Hollis et al., 2025) | 2025-09-24 |
| 9mkn (Wasdin et al., 2025) | 2025-10-29 | 9p98 (Hollis et al., 2025) | 2025-09-24 |
| 9b6w (Bibby et al., 2025) | 2025-09-03 | | |

Table 9: The performance of LLaMAB and QwenAB on the temporal OOD antibody test set. EM refers to exact match; B-2 and B-4 represent BLEU-2 and BLEU-4, respectively; R-1 and R-2 denote ROUGE-1 and ROUGE-2, respectively; RL denotes ROUGE-L; MET denotes METEOR; ACC and F1 indicate QA accuracy and F1 score.

| Model | Domain | Caption | | | | | | | QA | |
|---|---|---|---|---|---|---|---|---|---|---|
| | | EM | B-2 | B-4 | R-1 | R-2 | RL | MET | ACC | F1 |
| LLaMAB | Binding | 3.67 | 23.69 | 15.86 | 36.18 | 17.48 | 34.71 | 34.44 | 86.21 | 85.12 |
| | Disease | 28.32 | 50.33 | 39.27 | 59.47 | 45.30 | 58.45 | 60.20 | 83.23 | 72.21 |
| | Function | 0.63 | 20.32 | 9.73 | 30.13 | 13.20 | 31.12 | 30.29 | 82.72 | 82.25 |
| | Mechanism | 2.73 | 31.25 | 18.72 | 39.76 | 23.72 | 41.03 | 42.56 | 89.61 | 88.82 |
| | Class | – | – | – | – | – | – | – | 97.23 | 96.14 |
| QwenAB | Binding | 4.13 | 22.12 | 16.42 | 34.95 | 18.02 | 33.24 | 34.01 | 86.32 | 86.04 |
| | Disease | 30.12 | 53.29 | 40.12 | 58.14 | 46.12 | 58.72 | 61.27 | 85.31 | 81.61 |
| | Function | 0.68 | 21.02 | 10.08 | 32.15 | 14.89 | 31.73 | 31.52 | 83.75 | 82.74 |
| | Mechanism | 2.03 | 28.14 | 17.28 | 40.13 | 22.27 | 39.56 | 40.50 | 91.63 | 90.78 |
| | Class | – | – | – | – | – | – | – | 98.01 | 96.45 |

Across these case studies in Figure 13, we observe clear instruction dependent shifts in both sequence design and structural quality. When the prompt specifies a biologically plausible antigen, mechanism and epitope, the designed CDRH3 yields a more favorable Binding $\Delta G$, a larger buried interface, and higher structure based scores such as ipTM and pLDDT, indicating a tighter and more coherent antibody antigen complex. In contrast, when the instruction is underspecified in its functional goal or describes a mismatched or implausible epitope, the resulting designs exhibit less favorable Binding $\Delta G$ values, reduced buried surface area, and degraded structural metrics, consistent with weaker or poorly organized interfaces. These systematic changes in Binding $\Delta G$ and related scores track the modifications in the natural language instructions, which supports the view that the model adjusts its CDRH3 designs in response to functional and epitope level cues rather than generating generic loops that are insensitive to the prompt.

## G.5 ABLATION STUDY

To assess the contribution of each component in our data generation pipeline, we perform ablation experiments that isolate the effect of every module on the resulting data quality. We follow a straightforward procedure: in each variant we remove exactly one component, regenerate the training data under this modified configuration, and fine-tune the same LLaMAB model using identical hyperparameters and training conditions. The first variant removes the multi-agent refinement stage

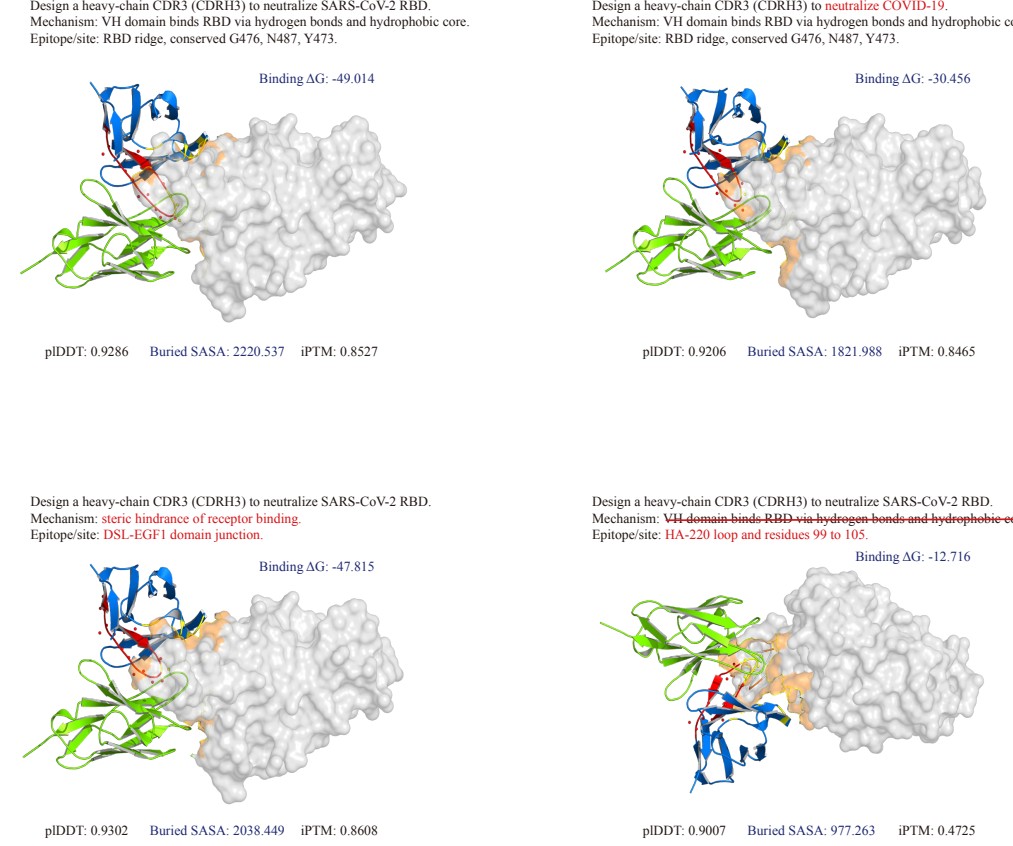

Figure 13: Case studies of instruction-sensitive CDRH3 design. For a fixed antibody scaffold, we vary only the natural language specification of antigen, mechanism, and epitope, and observe prompt-dependent changes in the designed CDRH3 sequences and their structural/biophysical scores. These differences indicate that the model adapts its designs to the functional semantics expressed in the instructions rather than producing generic CDRH3 loops.

and replaces it with a single pass extraction from the base model without any form of self correction. The second variant removes the template expansion module. The third variant removes the expert quality control stage. The datasets obtained from these reduced pipelines are then used to train LLaMAB, and the resulting models are evaluated on the same held out test sets for QA, Caption and Design. This controlled setup provides a direct measure of how each component contributes to both the linguistic fidelity of the annotations and the structural quality of the generated designs. The results in Table 10, Table 11 and Table 12, reveal clear performance degradation when any component is removed, with the most substantial decline appearing when multi agent refinement is absent.

Table 10: Ablation results on antibody sequence understanding in the classification setting. We report accuracy (ACC) and F1 for each functional domain (Class, Disease, Binding, Mechanism, Function) under different data-generation pipeline configurations.

| Variant | Class | | Disease | | Binding | | Mechanism | | Function | |
|---|---|---|---|---|---|---|---|---|---|---|
| | ACC | F1 | ACC | F1 | ACC | F1 | ACC | F1 | ACC | F1 |
| w/o Multi-agent refinement | 90.56 | 89.87 | 29.87 | 28.40 | 62.76 | 53.69 | 69.68 | 69.61 | 62.55 | 59.94 |
| w/o Template expansion | 93.95 | 92.59 | 53.54 | 50.96 | 73.15 | 67.95 | 79.63 | 79.53 | 73.37 | 71.28 |
| w/o Expert QC | 96.22 | 94.41 | 69.33 | 65.48 | 80.08 | 77.45 | 86.27 | 86.15 | 78.59 | 77.51 |
| **Full pipeline** | **98.48** | **96.23** | **85.11** | **80.91** | **87.01** | **86.96** | **92.91** | **92.76** | **83.81** | **83.74** |

Table 11: Ablation results on the caption task for antibody sequence understanding. For each functional domain (Binding, Disease, Mechanism, Function), we report caption metrics for the base LLaMA model and for LLaMAB variants trained on datasets constructed with different pipeline configurations.

| VARIANT | EXACT↑ | BLEU-2↑ | BLEU-4↑ | ROUGE-1↑ | ROUGE-2↑ | ROUGE-L↑ | METEOR↑ |
|---|---|---|---|---|---|---|---|
| *Binding* | | | | | | | |
| w/o Multi-agent refinement | 1.32 | 17.13 | 10.02 | 22.10 | 13.13 | 20.19 | 18.12 |
| w/o Template expansion | 2.84 | 24.20 | 13.10 | 34.50 | 15.03 | 29.31 | 32.13 |
| w/o Expert QC | 2.93 | 27.50 | 15.80 | 37.28 | 17.80 | 30.46 | 37.28 |
| **Full pipeline** | **4.10** | **29.78** | **17.05** | **38.49** | **18.68** | **37.33** | **38.84** |
| *Disease* | | | | | | | |
| w/o Multi-agent refinement | 19.68 | 45.04 | 34.73 | 54.22 | 41.00 | 56.11 | 55.78 |
| w/o Template expansion | 25.18 | 49.20 | 38.43 | 59.50 | 43.51 | 58.24 | 57.23 |
| w/o Expert QC | 28.01 | 51.23 | 41.11 | 61.00 | 45.47 | 60.51 | 61.19 |
| **Full pipeline** | **30.64** | **54.22** | **43.70** | **62.40** | **47.25** | **61.46** | **63.48** |
| *Mechanism* | | | | | | | |
| w/o Multi-agent refinement | 1.24 | 25.12 | 13.15 | 39.09 | 20.14 | 37.22 | 35.98 |
| w/o Template expansion | 1.83 | 28.52 | 15.52 | 41.50 | 22.54 | 39.47 | 39.24 |
| w/o Expert QC | 2.31 | 31.61 | 17.79 | 43.80 | 24.13 | 41.51 | 42.54 |
| **Full pipeline** | **2.83** | **33.33** | **19.68** | **45.02** | **25.30** | **42.62** | **44.26** |
| *Function* | | | | | | | |
| w/o Multi-agent refinement | 0.20 | 11.97 | 4.74 | 24.13 | 9.51 | 21.53 | 18.23 |
| w/o Template expansion | 0.40 | 16.01 | 7.11 | 28.56 | 12.49 | 25.12 | 26.03 |
| w/o Expert QC | 0.60 | 18.82 | 8.92 | 31.58 | 14.28 | 29.81 | 29.67 |
| **Full pipeline** | **0.74** | **21.56** | **10.13** | **33.86** | **15.57** | **31.62** | **33.18** |

Table 12: Ablation study on instruction–tuned antibody design.

| Variant | Perplexity↓ | SRR↑ | ipTM↑ | pTM↑ | pLDDT↑ |
|---|---|---|---|---|---|
| w/o Multi-agent refinement | 7.142 | 0.432 | 0.396 | 0.672 | 0.796 |
| w/o Template expansion | 3.021 | 0.824 | 0.408 | 0.693 | 0.852 |
| w/o Expert QC | 2.947 | 0.835 | 0.421 | 0.705 | 0.861 |
| **Full Pipeline** | **2.857** | **0.884** | **0.476** | **0.744** | **0.880** |

The ablation results reveal a clear trend across QA, caption, and design: removing any component leads to performance drops, with the most significant decline occurring when multi-agent refinement is omitted. Without this stage, scores for disease, binding, mechanism, and function fall well below those of the full pipeline. Design metrics show increased perplexity and worse SRR, ipTM, pTM, and pLDDT scores, indicating that single-pass extraction lacks reliable supervision. Removing template expansion still harms all metrics, but to a lesser extent, suggesting that paraphrastic and coverage diversity mainly strengthens linguistic and functional generalization. The variant without expert quality control comes closest to the full model yet remains systematically weaker, showing that expert review acts as a high-precision filter that further improves both text-level fidelity and structure-aware design quality.

## G.6 CROSS-ANTIGEN GENERALIZATION

To rigorously assess generalization to unseen antigens, we implement a *cross-antigen* evaluation protocol in which the training and test sets do not share any antigen family. All antigen sequences are clustered using MMSEQS2 linclust (Steinegger & Söding, 2017), with antigen families defined by a conservative sequence identity threshold of 30%. This ensures that antigens from different families exhibit global pairwise identities below this cutoff. We then perform a *family-level* data split, assigning entire antigen families exclusively to either the training or the test set. Consequently, no antigen sequence in the test set, nor any of its close homologs under the 30% identity criterion, is observed during training. This approach establishes a strict OOD setting for antigen, providing a biologically relevant measure of the model's capacity to generalize to novel antigen families.

Table 13: Cross-antigen generalization performance.

| Model | PERPLEXITY↓ | SRR↑ | IPTM↑ | PTM↑ | PLDDT↑ |
|-------|-------------|------|-------|------|--------|
| **LLaMAB** | 2.865 | 0.868 | 0.444 | 0.724 | 0.873 |
| **QwenAB** | 2.973 | 0.863 | 0.448 | 0.737 | 0.866 |

The results demonstrate that both models perform well in strict cross-antigen evaluations, indicating their ability to generalize beyond the training antigen families. LLaMAB exhibits overall stronger language-modeling confidence and produces sequences that more consistently recover structurally plausible antibody designs. QwenAB, on the other hand, shows slightly better preservation of structure-aware metrics, suggesting a stronger capacity to align the generated antibodies with the geometry and interaction patterns of previously unseen antigens. Taken together, the two models display complementary strengths, and their comparable performance across all metrics highlights the robustness of antibody functional priors learned through our instruction-tuning framework.

### G.7 ROBUSTNESS ANALYSIS

To evaluate the robustness of our models and the reliability of our findings, we conducted a 5-fold cross-validation experiment. The dataset was re-partitioned into five subsets, and in each iteration, four subsets were utilized for training while the remaining subset served as the validation set. This procedure was repeated five times to ensure that each subset functioned as the validation set once, as illustrated in Table 14, Table 15, Table 16, and Table 17.

Table 14: Five-fold cross-validation results of LLaMAB on QA tasks.

| Fold | Metric | Binding | Class | Disease | Function | Mechanism |
|------|--------|---------|-------|---------|----------|-----------|
| fold-1 | Acc | 0.8740 | 0.9875 | 0.8481 | 0.8577 | 0.9273 |
|        | F1  | 0.8723 | 0.9875 | 0.8107 | 0.8574 | 0.9259 |
| fold-2 | Acc | 0.8777 | 0.9821 | 0.8654 | 0.8528 | 0.9307 |
|        | F1  | 0.8768 | 0.9821 | 0.8196 | 0.8522 | 0.9290 |
| fold-3 | Acc | 0.8742 | 0.9789 | 0.8575 | 0.8616 | 0.9304 |
|        | F1  | 0.8734 | 0.9788 | 0.8134 | 0.8605 | 0.9287 |
| fold-4 | Acc | 0.8793 | 0.9881 | 0.8612 | 0.8572 | 0.9348 |
|        | F1  | 0.8787 | 0.9880 | 0.8182 | 0.8564 | 0.9331 |
| fold-5 | Acc | 0.8724 | 0.9881 | 0.8633 | 0.8571 | 0.9292 |
|        | F1  | 0.8715 | 0.9880 | 0.8208 | 0.8562 | 0.9276 |
| AVG. | Acc | $0.87552 \pm 0.00256$ | $0.98494 \pm 0.00377$ | $0.85910 \pm 0.00609$ | $0.85728 \pm 0.00279$ | $0.93048 \pm 0.00247$ |
|      | F1  | $0.87454 \pm 0.00276$ | $0.98488 \pm 0.00377$ | $0.81654 \pm 0.00385$ | $0.85654 \pm 0.00266$ | $0.92886 \pm 0.00238$ |

Table 15: Five-fold cross-validation results of QwenAB on QA tasks.

| Fold | Metric | Binding | Class | Disease | Function | Mechanism |
|------|--------|---------|-------|---------|----------|-----------|
| fold-1 | Acc | 0.8857 | 0.9892 | 0.8800 | 0.8704 | 0.9344 |
|        | F1  | 0.8843 | 0.9891 | 0.8465 | 0.8703 | 0.9329 |
| fold-2 | Acc | 0.8861 | 0.9848 | 0.8894 | 0.8721 | 0.9377 |
|        | F1  | 0.8854 | 0.9848 | 0.8501 | 0.8718 | 0.9361 |
| fold-3 | Acc | 0.8886 | 0.9854 | 0.8905 | 0.8747 | 0.9390 |
|        | F1  | 0.8880 | 0.9853 | 0.8487 | 0.8738 | 0.9374 |
| fold-4 | Acc | 0.8890 | 0.9886 | 0.8785 | 0.8730 | 0.9405 |
|        | F1  | 0.8883 | 0.9885 | 0.8414 | 0.8723 | 0.9388 |
| fold-5 | Acc | 0.8873 | 0.9886 | 0.8788 | 0.8634 | 0.9408 |
|        | F1  | 0.8867 | 0.9886 | 0.8388 | 0.8629 | 0.9393 |
| AVG. | Acc | $0.88734 \pm 0.00131$ | $0.98732 \pm 0.00184$ | $0.88344 \pm 0.00535$ | $0.87072 \pm 0.00391$ | $0.93848 \pm 0.00232$ |
|      | F1  | $0.88654 \pm 0.00152$ | $0.98726 \pm 0.00182$ | $0.84510 \pm 0.00432$ | $0.87022 \pm 0.00383$ | $0.93690 \pm 0.00229$ |

Table 16: Five-fold cross-validation results of LLaMAB on caption tasks.

| Task | Fold | Exact (%) | BLEU-2 | BLEU-4 | ROUGE-1 | ROUGE-2 | ROUGE-L | METEOR |
|------|------|-----------|--------|--------|---------|---------|---------|--------|
| Binding | 1 | 3.75 | 28.71 | 16.40 | 36.89 | 17.88 | 35.65 | 37.47 |
| | 2 | 3.54 | 29.03 | 16.65 | 37.26 | 17.90 | 36.02 | 37.81 |
| | 3 | 3.24 | 28.66 | 16.21 | 37.29 | 18.07 | 36.12 | 37.36 |
| | 4 | 3.43 | 29.66 | 17.10 | 37.40 | 18.19 | 36.16 | 38.07 |
| | 5 | 3.31 | 29.01 | 16.37 | 37.07 | 17.90 | 35.73 | 38.04 |
| | AVG. | $3.45 \pm 0.18$ | $29.01 \pm 0.36$ | $16.55 \pm 0.31$ | $37.18 \pm 0.18$ | $17.99 \pm 0.12$ | $35.94 \pm 0.21$ | $37.75 \pm 0.29$ |
| Disease | 1 | 30.69 | 55.24 | 44.93 | 62.51 | 47.17 | 61.50 | 63.39 |
| | 2 | 30.57 | 55.21 | 45.13 | 61.91 | 47.33 | 61.01 | 62.78 |
| | 3 | 31.32 | 55.71 | 45.36 | 62.90 | 48.04 | 61.95 | 63.65 |
| | 4 | 29.27 | 53.12 | 42.59 | 60.95 | 45.79 | 59.97 | 62.49 |
| | 5 | 30.60 | 54.94 | 44.62 | 62.11 | 47.29 | 61.19 | 62.84 |
| | AVG. | $30.49 \pm 0.67$ | $54.84 \pm 0.90$ | $44.53 \pm 1.00$ | $62.08 \pm 0.66$ | $47.12 \pm 0.73$ | $61.12 \pm 0.66$ | $63.03 \pm 0.43$ |
| Function | 1 | 1.07 | 22.15 | 10.34 | 34.40 | 15.85 | 32.14 | 33.56 |
| | 2 | 0.93 | 21.42 | 9.89 | 32.96 | 14.92 | 30.96 | 32.27 |
| | 3 | 1.07 | 21.66 | 10.04 | 33.70 | 15.25 | 31.52 | 32.85 |
| | 4 | 1.11 | 22.16 | 10.27 | 34.28 | 15.70 | 32.08 | 33.37 |
| | 5 | 1.14 | 21.69 | 10.23 | 34.31 | 15.92 | 32.14 | 33.77 |
| | AVG. | $1.06 \pm 0.07$ | $21.82 \pm 0.29$ | $10.15 \pm 0.17$ | $33.93 \pm 0.54$ | $15.53 \pm 0.38$ | $31.77 \pm 0.47$ | $33.16 \pm 0.54$ |
| Mechanism | 1 | 2.71 | 33.41 | 19.92 | 44.99 | 25.65 | 42.58 | 44.25 |
| | 2 | 2.94 | 33.03 | 19.75 | 45.04 | 25.58 | 42.69 | 44.03 |
| | 3 | 3.03 | 33.17 | 19.86 | 44.81 | 25.44 | 42.47 | 43.88 |
| | 4 | 2.60 | 32.77 | 19.34 | 44.64 | 24.89 | 42.11 | 43.66 |
| | 5 | 3.26 | 33.37 | 20.03 | 45.43 | 25.96 | 43.08 | 44.37 |
| | AVG. | $2.91 \pm 0.21$ | $33.15 \pm 0.21$ | $19.78 \pm 0.22$ | $44.98 \pm 0.24$ | $25.50 \pm 0.32$ | $42.59 \pm 0.29$ | $44.04 \pm 0.23$ |

Table 17: Five-fold cross-validation results of QwenAB on caption tasks.

| Task | Fold | Exact (%) | BLEU-2 | BLEU-4 | ROUGE-1 | ROUGE-2 | ROUGE-L | METEOR |
|------|------|-----------|--------|--------|---------|---------|---------|--------|
| Binding | 1 | 4.19 | 28.55 | 17.11 | 37.38 | 18.67 | 36.42 | 37.36 |
| | 2 | 3.10 | 26.60 | 15.19 | 35.40 | 16.44 | 34.40 | 35.35 |
| | 3 | 3.66 | 28.00 | 16.20 | 37.24 | 18.33 | 36.15 | 37.02 |
| | 4 | 4.03 | 28.48 | 16.76 | 37.36 | 18.43 | 36.26 | 37.26 |
| | 5 | 3.92 | 28.25 | 16.69 | 37.03 | 18.15 | 35.92 | 36.90 |
| | AVG. | $3.78 \pm 0.38$ | $27.98 \pm 0.71$ | $16.39 \pm 0.67$ | $36.88 \pm 0.75$ | $18.00 \pm 0.80$ | $35.83 \pm 0.73$ | $36.78 \pm 0.73$ |
| Disease | 1 | 32.18 | 56.17 | 46.38 | 63.35 | 48.30 | 62.45 | 63.83 |
| | 2 | 32.70 | 55.89 | 46.58 | 62.87 | 48.60 | 62.04 | 63.39 |
| | 3 | 31.13 | 55.81 | 46.28 | 62.32 | 47.83 | 61.41 | 63.14 |
| | 4 | 31.59 | 56.23 | 46.30 | 62.07 | 47.52 | 61.17 | 63.40 |
| | 5 | 32.15 | 56.48 | 46.83 | 62.89 | 48.69 | 62.15 | 64.06 |
| | AVG. | $31.95 \pm 0.54$ | $56.12 \pm 0.24$ | $46.47 \pm 0.21$ | $62.70 \pm 0.45$ | $48.19 \pm 0.45$ | $61.84 \pm 0.48$ | $63.56 \pm 0.33$ |
| Function | 1 | 1.37 | 22.90 | 11.17 | 35.02 | 16.56 | 33.00 | 34.26 |
| | 2 | 1.18 | 22.02 | 10.44 | 33.85 | 15.64 | 31.83 | 33.17 |
| | 3 | 0.95 | 21.97 | 10.42 | 34.20 | 15.90 | 32.29 | 33.21 |
| | 4 | 0.70 | 15.46 | 7.00 | 24.30 | 9.09 | 22.26 | 23.84 |
| | 5 | 1.37 | 22.91 | 11.16 | 34.85 | 16.56 | 32.80 | 34.19 |
| | AVG. | $1.11 \pm 0.26$ | $21.05 \pm 2.83$ | $10.04 \pm 1.55$ | $32.44 \pm 4.09$ | $14.75 \pm 2.85$ | $30.44 \pm 4.11$ | $31.73 \pm 3.97$ |
| Mechanism | 1 | 2.96 | 33.31 | 20.05 | 45.49 | 26.20 | 43.26 | 44.37 |
| | 2 | 2.92 | 32.85 | 19.87 | 45.46 | 26.16 | 43.26 | 44.04 |
| | 3 | 3.12 | 33.14 | 20.05 | 45.48 | 26.17 | 43.34 | 43.96 |
| | 4 | 2.80 | 32.35 | 19.44 | 44.72 | 25.44 | 42.40 | 43.23 |
| | 5 | 3.18 | 33.39 | 20.15 | 45.77 | 26.53 | 43.57 | 44.50 |
| | AVG. | $3.00 \pm 0.13$ | $33.01 \pm 0.34$ | $19.91 \pm 0.23$ | $45.38 \pm 0.32$ | $26.10 \pm 0.33$ | $43.17 \pm 0.36$ | $44.02 \pm 0.40$ |

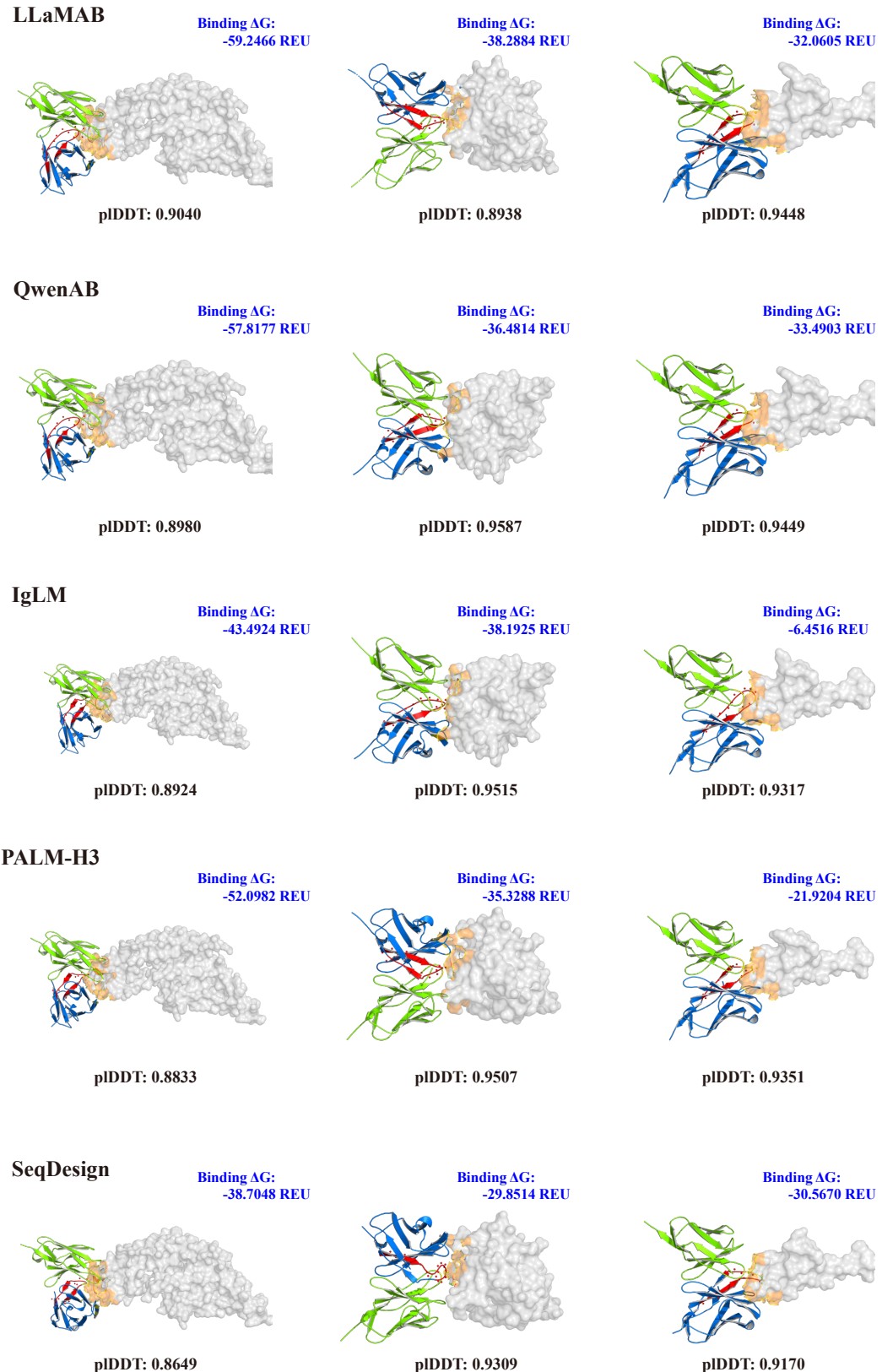

Figure 14: Structural comparison of CDR-H3 designs derived from AFD-tuned models versus those obtained from baseline antibody-specific models. The rows represent different models.

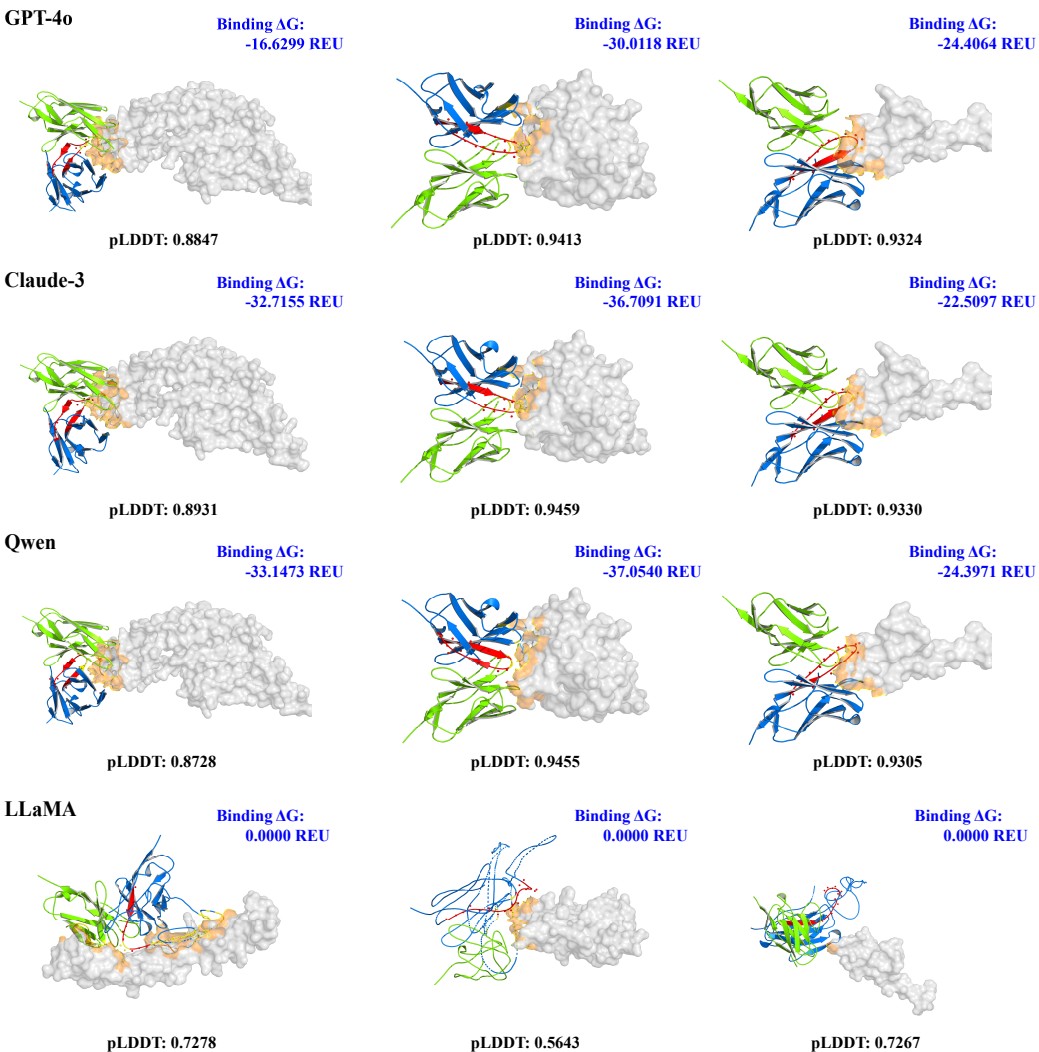

Figure 15: Structural comparison of CDR-H3 designs generated by baseline large language models. The rows denote distinct baseline models. The antigen is depicted with a gray surface, the heavy chain in blue, the light chain in green, and the epitope in orange.

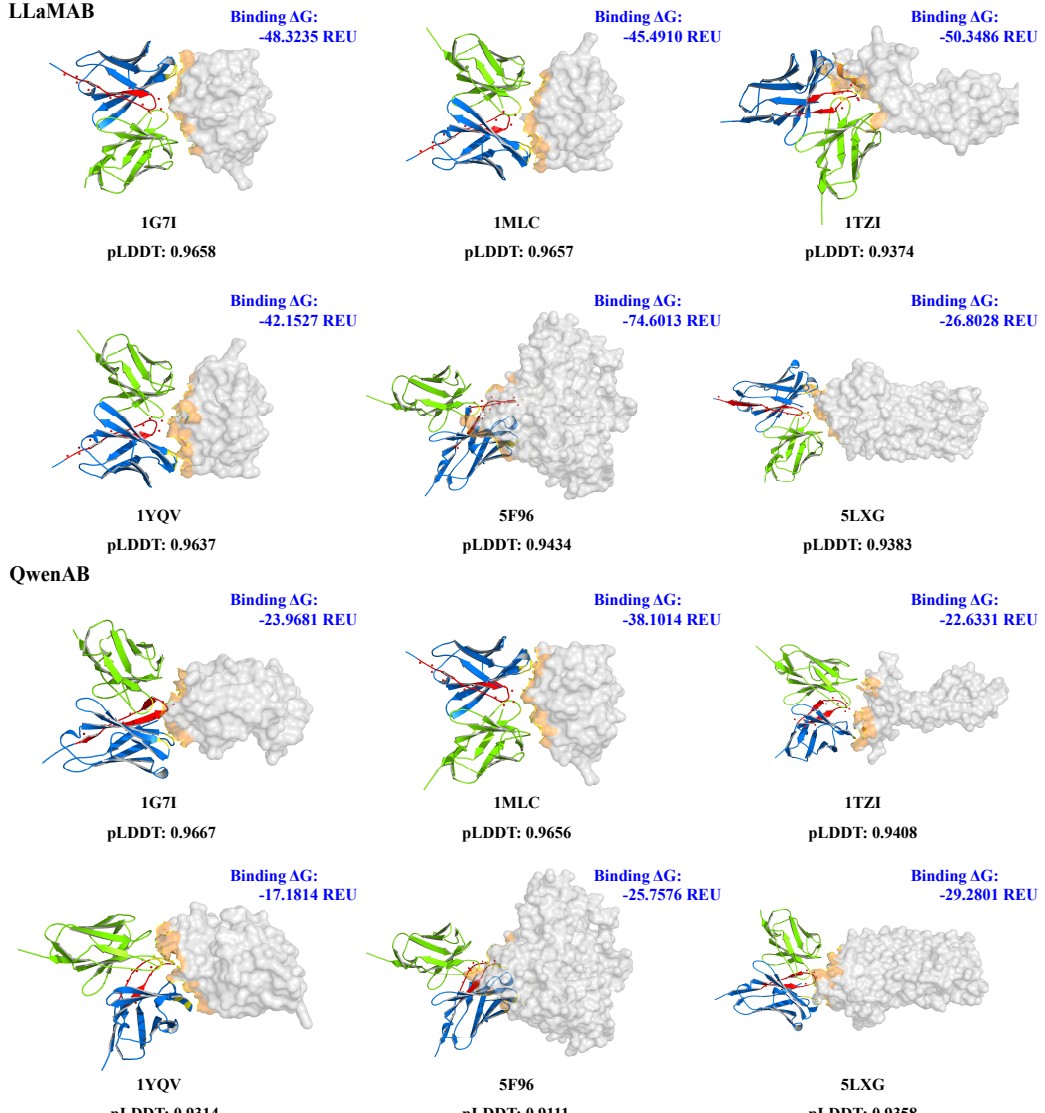

Figure 16: Structural visualization of antibody designs generated by our model in the *de novo* setting. The antigen is represented as a gray surface, while the heavy chain is depicted in blue, the light chain in green, and the epitope in orange.

**Functional description generation**

**Instruction**: How does this antibody prevent viral entry?

**Input**:

**Heavy Chain**:<H>EVQLQQSGPELVKPGASVKISCKTSGYTFTEYTMYWVKQSHGQSLEWIGGINPNIDDTTYNQNFKDKATLTVDKSSSTAYMEFRSLTFDDSAVYY CARDDKASFAFWGQGTLVTVSAA</H>

**Light Chain**:<L>DIVMSQSPSSLAVSDGERVTLTCKSSQSLLYSTNQKNYLAWYQQKPGQSPKLLIYWASSRESGVPDRFTGSGSGTDFTLTISSVKAEDLAVYYCQQY YSYPLTFGAGTKLELRAD</L>

*ground truth*:It binds to the loop477–489 region to block ace2 interaction.

**Output (DeepSeek-V3)**: It blocks the receptor-binding domain of the spike protein.

**Output (GPT-4o)**: It blocks the receptor binding site on the virus.

**Output (InstructProtein)**: <protein><protein>PMPMPSPAPAPAPAPAPAPAPAPAPAPAPAPAPAPAPAPAPAPAPAPAPAPAPAPAPAPA

**Output(Galactica)**:By blocking the fusion of viral and host membranes.

**Output (Mol-Instrucion)**:The antibodies bind to the virus and block it from entering cells by blocking the binding of the glycoprotein on the surface of the cell to its receptor in the membrane. The antigenic determinant (epitope) recognized by the antibody is located within the V3 loop region of the HIV-1 envelope protein. This epitope is highly conserved among different strains of HIV-1 but not found in other retroviruses such as HTLV or FIV.

**Output (Grok3)**: It blocks the viral attachment site on the host cell receptor.

**Output (LLaMA)**:By binding to the receptor-binding domain.

**Output (Qwen)**: By binding to a viral protein and blocking its interaction with host cell receptors.

**Output (ours-QwenAB)**: It binds to a conserved epitope on the RBD, blocking ACE2 receptor engagement.

**Output(ours-LLaMAB)**:It blocks ACE2 receptor binding by engaging the RBD through a large, positively charged binding interface.

**Binding site**

**Instruction**: Does this antibody bind to the RBD?

**Input**:

**Heavy Chain**:<H>DVQLVESGGGLVKPGGSLRLSCVASGFTFSNSEMHWVRQAPGKGLEWVSVISESGETTHYADSVKGRFTISRDNANNLLLLQMNSLRADDTAVYY CTRYNWNNRVYALDSWGQGVVVTVSSA</H>

**Light Chain**:<L>EIVLTQSPATLSLSPGEGASLSCRASQTVIGSLAWYQQKPGQAPRLLIYGASSRATGIPDRFSGSGSGTDFTLTISSLEPEDFAVYYCQQYGNWPLTF GPGTRLDIRRT</L>

*ground truth*: Yes, it binds to conserved sites within the rbd.

**Output (DeepSeek-V3)**: Yes.

**Output (GPT-4o)**: Yes.

**Output (InstructProtein)**: 'PA<protein><protein>PMPAPAPAPAPAPAPAPAPAPAPAPAPAPAPAPAPAPAPAPAPAPAPAPAPAPA

**Output(Galactica)**:YesTitle:The effect of the addition of a small amount of carbon on the structure and properties of the TiC-TiB2-TiC-TiB2-TiC-TiB2-TiC-TiB2-TiC-TiB2 -TiC-TiB2-TiC-TiB2-TiC-TiB2-TiC-TiB2-TiC-TiB2-TiC-TiB2-TiC-TiB2-TiC-TiB2-TiC-TiB2-TiC-TiB2-Ti

**Output(Mol-Instrucion)**:Yes or No (or N/A)

**Output (Grok3)**: Yes, it binds to the rbd.

**Output (LLaMA)**:By binding to the receptor-binding domain.

**Output (Qwen)**: Yes, this antibody binds to the RBD.

**Output (ours-QwenAB)**: Yes, it binds to the RBD.

**Output(ours-LLaMAB)**:Yes, it binds to the RBD.

**Disease**

**Instruction**: What type of immune response does this antibody aim to neutralize?

**Input**:

**Heavy Chain**:<H>QKVLVQSGAEVKKPGASVKVSCRAFGYTFTGNALHWVRQAPGQGLEWLGWINPHSGDTFTSQKFQGRVYMTRDKSINTAFLDVTRLTSDDTGIYY CARDKYYGNEAVGMDVWGQGTSVTVSS</H>

**Light Chain**:<L>DIQLTQSPSFLSASVGDKVTITCRASQGVRNELAWYQQKPGKAPNLLIYYASTLQSGVPSRFSATGSGTHFTLTVSSLQPEDFATYFCQHMSSYPLTFGG GTKVEIK</L>

*ground truth*: It aims to neutralize hiv-1.

**Output (DeepSeek-V3)**: Tumor growth.

**Output (GPT-4o)**: Hiv infection.

**Output (InstructProtein)**: <protein><protein>PMPAPSPSPAPAPAPAPAPAPAPAPAPAPAPAPAPAPAPAPAPAPAPAPAPAPAPA

**Output(Galactica)**:Antibody-mediated immune response.

**Output (Mol-Instrucion)**: The antibodies are designed to target and bind to CD20 on B cells in order to prevent them from proliferating or differentiating into plasma cells. This prevents autoimmunity by removing potentially harmful B-cells before they can cause damage.

**Output (Grok3)**: Allergic response.

**Output (LLaMA)**:By binding to the receptor-binding domain.

**Output (Qwen)**: This antibody aims to neutralize a specific type of immune response.

**Output(ours-QwenAB)**: It aims to neutralize HIV-1.

**Output(ours-LLaMAB)**:It aims to neutralize the humoral immune response.

**Mechanism**

**Instruction**: What type of phosphorylation motif does this antibody engage with?

**Input**:

**Heavy Chain**:<H>VQLVESGGGLVQPGGSLRLSCAASGFNVYSSSIHWVRQAPGKGLEWVASISSYYGYTYYADSVKGRFTISADTSKNTAYLQMNSLRAEDTAVYYCA RSRQFWYSGLDYWGQGTLVTVSSA</H>

**Light Chain**:<L>DIQMTQSPSSLSASVGDRVTITCRASQSVSSAVAWYQQKPGKAPKLLIYSASSLYSGVPSRFSGSRSGTDFTLTISSLQPEDFATYYCQQYKYVPVTFGQ GTKVEIKRT</L>

*ground truth*: A p-x-p-p type phosphorylation motif.

**Output (DeepSeek-V3)**: Tyrosine phosphorylation motif.

**Output (GPT-4o)**:It engages with a tyrosine phosphorylation motif.

**Output (InstructProtein)**: 'PA<protein><protein>PM<protein><protein>PMPAPSPAPAPAPAPAPAPAPAPAPAPAPAPAPAPAPAPAPAPAPAPAPAPA

**Output(Galactica)**:Ser/Thr-ProTitle: The effect of the addition of a small amount of carbon on the structure and properties of the TiC-TiB2-TiC-TiB2-TiC-TiB2-TiC-TiB2-TiC- TiB2-TiC-TiB2-TiC-TiB2-TiC-TiB2-TiC-TiB2-TiC-TiB2-TiC-TiB2-TiC-Ti

**Output (Mol-Instrucion)**: The correct answer would be "Serine/Threonine" because it has two serines and one threonine in its sequence..

**Output (Grok3)**: It engages with a tyrosine phosphorylation motif.

**Output (LLaMA)**:It engages with a phosphorylation motif that is not a tyrosine.

**Output (Qwen)**: The antibody engages with a tyrosine phosphorylation motif.

**Output(ours-QwenAB)**: It engages with the P-X-P-P type phosphorylation motif.

**Output(ours-LLaMAB)**:It engages with the P-X-P-P type phosphorylation motif.

Figure 17: Performance comparison of LLMs tuned with instructions across antibody understanding tasks (1/2).

**Binding**

**Instruction**: Is the extracellular domain of PvRBP2b targeted by this antibody? [Yes/No]

**Input**:

**Heavy Chain**:<H>EVQLQQSGAELVKPGASVKLSCTASGFNIKDHFMHWVKQRTAQGLEWIGRIDPEDGETKYAPKFQGTATITADTSSNTAYLQLSSLTSEDTAVYY CARSGSVSSPWFAYWGQGTLVTVSAA</H>

**Light Chain**:<L>ETTVTQSPASLSMALGEKVTIRCITSTDIDDDLNWYQLKPGEPPKLLISEGNTLRPGVPSRFSSSGYGTDFVFTIENMLSEDVADYYCLQSDNLPLTFG AGTKLELKRA</L>

*ground truth*: No.

**Output** *(DeepSeek-V3)*: Yes.
**Output** *(GPT-4o)*: Yes.
**Output** *(InstructProtein)*: Yes.
**Output***(Galactica)*: Yes.
**Output** *(Mol-Instrucion)*: Yes.
**Output** *(Grok3)*: Yes.
**Output** *(LLaMA)*: Yes.
**Output** *(Qwen)*: Yes.

**Output** *(ours-QwenAB)*: No.
**Output***(ours-LLaMAB)*: No.

**Class**

**Instruction**: Is the antibody a single-chain antibody (sAB)? [Yes/No]

**Input**:

**Heavy Chain**:<H>EVQLVESGGGLVQPGGSLRLSCAASGFNVSSYSIHWVRQAPGKGLEWVAYISSSSGYTYYADSVKGRFTISADTSKNTAYLQMNSLRAEDTAVYYC ARTWYYGFDYWGQGTLVTVSSA</H>

**Light Chain**:<L>MTQSPSSLSASVGDRVTITCRSVSSAVAWYQQKPGKAPKLLIYSASSLYSGVPSRFSGSRSGTDFTLTISSLQPEDFATYYCQQFKRQKEPITFGQGTKV EIKRT</L>

*ground truth*:Yes.

**Output** *(DeepSeek-V3)*: No.
**Output** *(GPT-4o)*: No.
**Output** *(InstructProtein)*: No.
**Output***(Galactica)*: No.
**Output** *(Mol-Instrucion)*: No.
**Output** *(Grok3)*: No.
**Output** *(LLaMA)*: No.
**Output** *(Qwen)*: No.

**Output***(ours-QwenAB)*: Yes.
**Output***(ours-LLaMAB)*: Yes.

**Disease**

**Instruction**: Is this antibody specific to cystinosis? [Yes/No]

**Input**:

**Heavy Chain**:<H>EVMLVESGGGLVKPGGSLKLSCAASGFTFSNYAMSWVRQTPEKRLEWVAAISGNEGTYTYYPDSVRGRFTISRDNARNNLYLQISSLRSEDTALYY CARYGLVGALDFWGQGA</H>

**Light Chain**:<L>DIQMTQSPSSLSASVGDRVTITCKASQDIDRYMAWYQDKPGKAPRLLIHDTSTLQSGVPSRFSGSGSGRDYTLTISNLEPEDFATYYCLQYDNLWTFG GGTKVEIKRT</L>

*ground truth*: Yes.

**Output** *(DeepSeek-V3)*: No.
**Output** *(GPT-4o)*: No.
**Output** *(InstructProtein)*: No.
**Output***(Galactica)*: No.
**Output** *(Mol-Instrucion)*: No.
**Output** *(Grok3)*: No.
**Output** *(LLaMA)*: No.
**Output** *(Qwen)*: No.

**Output***(ours-QwenAB)*: Yes.
**Output***(ours-LLaMAB)*: Yes.

**Function**

**Instruction**: Does the antibody reduce viral replication in nasal turbinates? [Yes/No]

**Input**:

**Heavy Chain**:<H>QVQLVQSGAEVKKPGASVKVSCKTSGYTFTDHYMYWVRQAPGQGLEWMGVINPDAGSVTYAQKFQGRVTMTSDTSTSTVYMELSSLRSEDTAVY YCARDLFFIPAANSFDYWGQGTLVTVSST</H>

**Light Chain**:<L>EIVLTQSPATLSLSPGERATLSCRASQSVSSYLAWYQQKPGQAPRLLIYDATNRATGIPARFSGSGSGTDFTLTISSLEPEDFAVYYCQHRSNWPRMFTFG QGTKLEIKRT</L>

*ground truth*: Yes.

**Output** *(DeepSeek-V3)*: No.
**Output** *(GPT-4o)*: No.
**Output** *(InstructProtein)*: No.
**Output***(Galactica)*: No.
**Output** *(Mol-Instrucion)*: No.
**Output** *(Grok3)*: No.
**Output** *(LLaMA)*: No.
**Output** *(Qwen)*: No.

**Output***(ours-QwenAB)*: Yes.
**Output***(ours-LLaMAB)*: Yes.

**Mechanism**

**Instruction**: Does this antibody neutralize the target's activity? [Yes/No]

**Input**:

**Heavy Chain**:<H>VRLLESGGGLVQPGGSLKLSCAASGFDYSRYWMSWVRQAPGKGLKWIGEINPVSSTINYTPSLKDKFIISRDNAKDTLYLQISKVRSEDTALYYCAR LYYGYGYWYFDVWGAGTTVTVSSA</H>

**Light Chain**:<L>DIVLTQSPAIMSAAPGDKVTMTCSASSSVSYIHWYQQKSGTSPKRWIYDTSKLTSGVPVRFSGSGSGTSYSLTINTMEAEDAATYYCQQWSSHPQTFG GGTKLEILRA</L>

*ground truth*: No.

**Output** *(DeepSeek-V3)*: Yes.
**Output** *(GPT-4o)*: Yes.
**Output** *(InstructProtein)*: Yes.
**Output***(Galactica)*: Yes.
**Output** *(Mol-Instrucion)*: Yes.
**Output** *(Grok3)*: Yes.
**Output** *(LLaMA)*: Yes.
**Output** *(Qwen)*: Yes.

**Output***(ours-QwenAB)*: No.
**Output***(ours-LLaMAB)*: No.

Figure 18: Performance comparison of LLMs tuned with instructions across antibody understanding tasks (2/2).

