# OpenReview forum: "AFD-INSTRUCTION: A Comprehensive Antibody Instruction Dataset with Functional Annotations for LLM-Based Understanding and Design"
_ICLR.cc/2026/Conference — ICLR 2026 Conference Withdrawn Submission_

### Official Review · Reviewer_KkCC · 2025-10-29

**Soundness:** 3
**Presentation:** 3
**Contribution:** 3
**Rating:** 6
**Confidence:** 4

**Summary:**

This paper introduces AFD-Instruction, the first large-scale instruction dataset that links antibody sequences with natural-language functional annotations. The dataset is built through a multi-agent pipeline combining extraction, mechanistic reasoning, and expert verification. It supports two domains: antibody understanding and function-guided antibody design. The authors fine-tune open-source foundation models (Qwen, LLaMA) and demonstrate substantial gains over both protein-language and general-purpose LLMs in understanding and design metrics.

**Strengths:**

1. High-quality domain focus. Antibody-specific dataset aligning sequences, functional text, and design tasks, and it is the first of its kind.
2. Comprehensive pipeline. Multi-agent extraction and self-questioning expansion show thoughtful integration of automation and expert verification.
3. Rich evaluation. Covers both understanding and design tasks, with structural and energetic validation using tFold + Rosetta.
4. Substantial empirical gains. Instruction-tuned QwenAB/LLaMAB outperforms general and domain-specific baselines across multiple metrics.
5. Ethical considerations (dual-use awareness, controlled access) are helpful and detailed.

**Weaknesses:**

### W1 Limited methodological novelty.
The dataset-generation procedure largely extends patterns from Mol-Instructions and Evola (instruction synthesis via LLM prompting + self-questioning). The multi-agent framing repackages established extract–verify–summarize steps; conceptual contribution is modest beyond domain adaptation.
### W2 Small biological diversity, large linguistic inflation.
The dataset derives 430 K instructions from only 4.3 K antibodies, meaning linguistic diversity far exceeds biological diversity. It's unclear whether model improvements stem from true functional grounding or synthetic linguistic variety.
### W3 Absence of mechanistic generalization analysis.
The paper shows performance gains but not whether models generalize to unseen antigen classes or novel mechanisms. Without a cross-target or cross-epitope evaluation, "understanding" may still be surface-level pattern matching.
### W4 Evaluation focuses on text similarity rather than scientific correctness.
BLEU/ROUGE metrics reward lexical overlap but not mechanistic accuracy. Expert or structure-aware evaluation (e.g., contact-map consistency, mutation-effect prediction) would be more meaningful.
### W5 Limited ablation and uncertainty quantification.
No breakdown of how each module (Extractor, Mechanism, Function, Self-Questioning) affects data quality. Single-run results in Table 1 and Table 2 without error bars obscure robustness.
### W6 Weak discussion of relation to prior instruction datasets.
Although Table 5 lists comparisons, the paper underplays key distinctions from Evola, which already support large-scale protein instruction tuning, and doesn't clearly justify why antibody specificity requires a new corpus instead of targeted filtering from Evola.

**Questions:**

If the author could provide clarification to the points mentioned in the weakness section and the following questions, I am happy to consider raising the score.
1. Can models trained on AFD-Instruction transfer to unseen antigen families or novel antibody mechanisms (e.g., pH-dependent binding, conformational epitopes)? How do they perform under cross-antigen split evaluation?
2. How much do the multi-agent refinement, template expansion, and expert QC respectively improve accuracy or diversity of annotations?
3. Given only 4 K unique antibodies, how do you ensure the 430 K instructions are not redundant paraphrases? Did you analyze instruction uniqueness or semantic clustering?
4. Beyond domain restriction, what conceptual advances justify AFD-Instruction as a new dataset family rather than a subset or extension of Evola?
5. Would it be feasible to include domain-expert evaluation of generated mechanism descriptions, or at least automated structure-text alignment (e.g., verifying residue mentions against structural contacts)?

---

> ### Author Response · Authors · 2025-11-20
> **Response Part 1**
>
> Dear Review KkCC,
>
> We sincerely appreciate your encouraging and carefully considered review. Your positive assessment and constructive feedback mean a great deal to us, and we respond to your comments point by point below.
>
> > **W1.**  Limited methodological novelty.
>
> We agree that our data construction pipeline follows the now-standard pattern of LLM-assisted instruction synthesis and self-questioning used in Mol-Instructions and Evola, and we do not claim methodological novelty at the level of the general instruction-generation paradigm. The multi-agent components in our system operationalize the familiar extract–verify–summarize steps and are intended as a practical engineering choice rather than the main conceptual contribution. Our focus is instead on defining an antibody-centered and structure-grounded instruction space that did not previously exist.
>
> AFD-Instruction organizes antibody understanding into five coordinated dimensions—class, binding site, mechanism, disease relevance, and function—and enforces that each antibody–antigen complex is annotated consistently across these perspectives in both question–answer and caption formats. Constructing this space requires explicit grounding of natural language in experimentally determined structures from PDB and SAbDab, including epitope regions, the geometric relationship between paratope and epitope, and clinically substantiated indications. The same functional descriptions are further reused as design specifications for generating new variable-region and CDR-H3 sequences, thereby unifying functional understanding and function-guided design within a single coherent framework. These are domain-specific conceptual design choices tailored to antibody biology, rather than a new instruction-synthesis algorithm.
>
> The conceptual contribution of AFD-Instruction therefore lies in enabling a new research capability. To our knowledge, it is the first large-scale resource that integrates structural, mechanistic, functional, and clinical perspectives of antibodies within a unified instruction space. This structure allows large language models to perform forms of reasoning and generation that earlier instruction corpora for small molecules or generic proteins did not support, including cross-antigen generalization, temporal evaluation on newly released antibodies, mechanism-aware sequence design, and interpretable functional rationalization.
>
> > **W2.**  Small biological diversity, large linguistic inflation.
>
> To address this, we performed a semantic analysis of the instruction space, which shows that model improvements stem from true functional grounding, rather than from synthetic linguistic variety. Our analysis is detailed in **Appendix C ("Instruction Diversity and Semantic Structure")** and visualized in **Figure 10**.
>
> We randomly sampled a subset of instructions from the complete corpus and utilized an MPNet-based sentence encoder within the Sentence-BERT to embed their full texts. Subsequently, we projected the resulting vectors using UMAP. The embedding space reveals numerous distinct and well-separated semantic regions instead of collapsing into a limited number of dense modes, indicating a broad and meaningful range of instructional intents. Employing KMeans with thirty clusters resulted in cluster sizes ranging from 170 to 971 instructions, with a median size of 562; furthermore, the entropy of the cluster size distribution is measured at 3.33 nats—very close to the theoretical maximum of approximately 3.40 nats for thirty equally sized clusters—demonstrating that no small subset of clusters dominates the corpus.
>
> A nearest-neighbor analysis performed on a randomly selected set comprising four thousand instructions from the entire dataset yielded a mean cosine similarity score of 0.83 and a median score of 0.84; notably, no pairs exceeded a similarity threshold of 0.95. This finding provides evidence against large-scale paraphrastic duplication within our data set. Additionally, instructions associated with identical antibodies exhibit an average cosine similarity score of 0.595, while those linked to different antibodies show significantly lower average similarities at 0.506. This gap indicates that the semantic variation in AFD-Instruction is organized around biologically grounded distinctions such as antigen identity, mechanism of action, and disease relevance, rather than arising from superficial linguistic rewording.
>
> The collective quantitative observations suggest that AFD-Instruction contains a rich and biologically grounded semantic diversity, rather than exhibiting synthetic linguistic redundancy.

---

> ### Author Response · Authors · 2025-11-20
> **Response Part 2**
>
> > **W3.**  Absence of mechanistic generalization analysis.
>
> To directly address the reviewer’s concern that our models may not generalize to unseen antigen classes or novel mechanisms, we introduce a cross-antigen evaluation protocol in which the training and test sets do not share any antigen family.
>
> All antigen sequences are clustered using MMseqs2 linclust, and antigen families are defined by a conservative sequence-identity threshold of 30\%, ensuring that antigens belonging to different families exhibit global pairwise identities below this cutoff. Subsequently, we perform a family-level split of the data, assigning entire antigen families exclusively to either the training set or the test set. As a result, no antigen sequence in the test split—and no close homologues under the 30\% identity criterion—are ever encountered during training. This protocol establishes a biologically meaningful out-of-distribution setting at the antigen level and enables us to assess whether the models genuinely generalize to novel antigen families and their associated mechanisms rather than merely relying on superficial pattern matching. The results of this cross-antigen evaluation are presented below:
>
> | **Model**   | Perplexity ↓ | SRR ↑ | ipTM ↑ | pTM ↑ | pLDDT ↑ |
> |-------------|--------------|-------|--------|-------|---------|
> | **LLaMAB**  | 2.865        | 0.868 | 0.444  | 0.724 | 0.873   |
> | **QwenAB**  | 2.973        | 0.863 | 0.448  | 0.737 | 0.866   |
>
> As demonstrated in the results, both AFD-tuned models (QwenAB and LLaMAB) exhibit robust performance under this antigen-disjoint split, indicating that they do not depend on antigen memorization. While their strengths manifest slightly differently across various metrics, both models consistently maintain sequence plausibility and structure-aware alignment when assessed on completely unseen antigen families. These findings suggest that AFD-Instruction–tuned models are capable of transferring their learned antibody-antigen interaction patterns to antigen families not encountered during training, thereby indicating a meaningful level of cross-antigen generalization.
>
> > **W4.**  Evaluation focuses on text similarity rather than scientific correctness.
>
> - We appreciate the reviewer’s suggestion and agree that lexical metrics alone do not fully capture mechanistic correctness. In addition to BLEU/ROUGE, we therefore conducted a small-scale expert evaluation of the LLaMAB-generated mechanism descriptions. We invited three experts in antibody engineering and structural immunology, and each expert independently evaluated 50 randomly sampled generations from LLaMAB. For every sample, the experts rated (on a 0–3 scale) (i) mechanistic accuracy, (ii) consistency with the known antibody–antigen structure, and (iii) mechanistic granularity, i.e., whether the description goes beyond generic “neutralization/binding” statements to include fine-grained or dynamic information such as specific residues, loops, conformational changes, or escape-related effects. We report the mean score across experts and samples as well as the proportion of generations receiving a score of at least 2 (“broadly correct or better”) for each dimension. For LLaMAB, the average overall expert score is 2.52 out of 3, indicating that, on average, the generated mechanism descriptions are judged to be between “broadly correct” and “highly accurate”, with non-trivial structural consistency and mechanistic detail rather than purely superficial labels. This expert study thus complements BLEU/ROUGE by directly probing mechanistic correctness and structure-grounded plausibility of the generated explanations.
>
> - The models fine-tuned on AFD-Instruction also perform strongly on our antibody design benchmarks, including under cross-antigen (**Appendix G.6**) and OOD (**Appendix G.3**) settings.  Their high structure-based scores indicate that AFD-tuned models can propose sequences that fold into structurally plausible antibodies and form reasonable interfaces with their target antigens. Although these metrics are orthogonal to text generation, they are consistent with the view that AFD-Instruction helps the model internalize structure- and mechanism-aware antibody–antigen interaction patterns, providing complementary evidence beyond purely lexical measures.

---

> ### Author Response · Authors · 2025-11-20
> **Response Part 3 (w5)**
>
> > **W5.**  Limited ablation and uncertainty quantification.
>
> - we conducted a dedicated series of ablation experiments to provide a clear breakdown of how each module in our data-generation pipeline influences the quality of the resulting annotations and model performance. For each ablation variant, we removed exactly one component, regenerated the training data under this modified configuration, and fine-tuned the same LLaMA model using identical hyperparameters and training settings. The resulting models were then evaluated on the same held-out test sets across QA, Caption, and Design tasks, allowing us to directly quantify the contribution of each module.
>
> Across all tasks, we observe consistent and interpretable trends: removing any single component leads to a measurable deterioration in data quality and downstream performance, with the largest degradation occurring when multi-agent refinement is removed. These results address the reviewer’s concern by providing explicit, module-level evidence of robustness and by showing that the full pipeline yields substantially higher linguistic fidelity and structural accuracy than any reduced variant. Please see the tables below for the complete ablation results.
>
> **Table1: Ablation study on Classification Task**
>
> | Variant                     | Class ACC | Class F1 | Disease ACC | Disease F1 | Binding ACC | Binding F1 | Mechanism ACC | Mechanism F1 | Function ACC | Function F1 |
> |:----------------------------|----------:|---------:|------------:|-----------:|------------:|-----------:|--------------:|-------------:|-------------:|------------:|
> | w/o Multi-agent refinement | 90.56     | 89.87    | 29.87       | 28.40      | 62.76       | 53.69      | 69.68         | 69.61        | 62.55        | 59.94       |
> | w/o Template expansion     | 93.95     | 92.59    | 53.54       | 50.96      | 73.15       | 67.95      | 79.63         | 79.53        | 73.37        | 71.28       |
> | w/o Expert QC              | 96.22     | 94.41    | 69.33       | 65.48      | 80.08       | 77.45      | 86.27         | 86.15        | 78.59        | 77.51       |
> | **Full pipeline**          | **98.48** | **96.23**| **85.11**   | **80.91**  | **87.01**   | **86.96**  | **92.91**     | **92.76**    | **83.81**    | **83.74**   |
>
> ---
>
> **Table2: Ablation study on Caption Task (Binding)**
>
> | Variant                    | Exact | BLEU-2 | BLEU-4 | ROUGE-1 | ROUGE-2 | ROUGE-L | METEOR |
> |:---------------------------|------:|-------:|-------:|--------:|--------:|--------:|-------:|
> | w/o Multi-agent refinement | 1.32  | 17.13  | 10.02  | 22.10   | 13.13   | 20.19   | 18.12  |
> | w/o Template expansion     | 2.84  | 24.20  | 13.10  | 34.50   | 15.03   | 29.31   | 32.13  |
> | w/o Expert QC              | 2.93  | 27.50  | 15.80  | 37.28   | 17.80   | 30.46   | 37.28  |
> | **Full pipeline**          | **4.10** | **29.78** | **17.05** | **38.49** | **18.68** | **37.33** | **38.84** |
>
> ---
>
> **Table3: Ablation study on Caption Task (Disease)**
>
> | Variant                    | Exact | BLEU-2 | BLEU-4 | ROUGE-1 | ROUGE-2 | ROUGE-L | METEOR |
> |:---------------------------|------:|-------:|-------:|--------:|--------:|--------:|-------:|
> | w/o Multi-agent refinement | 19.68 | 45.04  | 34.73  | 54.22   | 41.00   | 56.11   | 55.78  |
> | w/o Template expansion     | 25.18 | 49.20  | 38.43  | 59.50   | 43.51   | 58.24   | 57.23  |
> | w/o Expert QC              | 28.01 | 51.23  | 41.11  | 61.00   | 45.47   | 60.51   | 61.19  |
> | **Full pipeline**          | **30.64** | **54.22** | **43.70** | **62.40** | **47.25** | **61.46** | **63.48** |
>
> ---
>
> **Table4: Ablation study on Caption Task (Mechanism)**
>
> | Variant                    | Exact | BLEU-2 | BLEU-4 | ROUGE-1 | ROUGE-2 | ROUGE-L | METEOR |
> |:---------------------------|------:|-------:|-------:|--------:|--------:|--------:|-------:|
> | w/o Multi-agent refinement | 1.24  | 25.12  | 13.15  | 39.09   | 20.14   | 37.22   | 35.98  |
> | w/o Template expansion     | 1.83  | 28.52  | 15.52  | 41.50   | 22.54   | 39.47   | 39.24  |
> | w/o Expert QC              | 2.31  | 31.61  | 17.79  | 43.80   | 24.13   | 41.51   | 42.54  |
> | **Full pipeline**          | **2.83** | **33.33** | **19.68** | **45.02** | **25.30** | **42.62** | **44.26** |
>
> ---
>
> **Table5: Ablation study on Caption Task (Function)**
>
> | Variant                    | Exact | BLEU-2 | BLEU-4 | ROUGE-1 | ROUGE-2 | ROUGE-L | METEOR |
> |:---------------------------|------:|-------:|-------:|--------:|--------:|--------:|-------:|
> | w/o Multi-agent refinement | 0.20  | 11.97  | 4.74   | 24.13   | 9.51    | 21.53   | 18.23  |
> | w/o Template expansion     | 0.40  | 16.01  | 7.11   | 28.56   | 12.49   | 25.12   | 26.03  |
> | w/o Expert QC              | 0.60  | 18.82  | 8.92   | 31.58   | 14.28   | 29.81   | 29.67  |
> | **Full pipeline**          | **0.74** | **21.56** | **10.13** | **33.86** | **15.57** | **31.62** | **33.18** |
>
> ---

---

> ### Author Response · Authors · 2025-11-20
> **Response Part 4 (w5)**
>
> **Table6: Ablation study on antibody design**
>
> | Variant                    | Perplexity ↓ | SRR ↑ | ipTM ↑ | pTM ↑ | pLDDT ↑ |
> |:---------------------------|-------------:|------:|-------:|------:|--------:|
> | w/o Multi-agent refinement | 7.142        | 0.432 | 0.396  | 0.672 | 0.796   |
> | w/o Template expansion     | 3.021        | 0.824 | 0.408  | 0.693 | 0.852   |
> | w/o Expert QC              | 2.947        | 0.835 | 0.421  | 0.705 | 0.861   |
> | **Full Pipeline**          | **2.857**    | **0.884** | **0.476** | **0.744** | **0.880** |
>
>
> - To assess stability beyond single-run results, we newly perform five-fold cross-validation for both LLaMAB and QwenAB on all QA and caption tasks.
>
> **Table7: Five-fold cross-validation on QA tasks (LLaMAB and QwenAB)**
>
> | Model  | Metric | Binding                  | Class                     | Disease                   | Function                  | Mechanism                 |
> |--------|--------|--------------------------|---------------------------|---------------------------|---------------------------|---------------------------|
> | LLaMAB | Acc    | $0.87552 \pm 0.00256$    | $0.98494 \pm 0.00377$     | $0.85910 \pm 0.00609$     | $0.85728 \pm 0.00279$     | $0.93048 \pm 0.00247$     |
> | LLaMAB | F1     | $0.87454 \pm 0.00276$    | $0.98488 \pm 0.00377$     | $0.81654 \pm 0.00385$     | $0.85654 \pm 0.00266$     | $0.92886 \pm 0.00238$     |
> | QwenAB | Acc    | $0.88734 \pm 0.00131$    | $0.98732 \pm 0.00184$     | $0.88344 \pm 0.00535$     | $0.87072 \pm 0.00391$     | $0.93848 \pm 0.00232$     |
> | QwenAB | F1     | $0.88654 \pm 0.00152$    | $0.98726 \pm 0.00182$     | $0.84510 \pm 0.00432$     | $0.87022 \pm 0.00383$     | $0.93690 \pm 0.00229$     |
>
> **Table8: Five-fold cross-validation on caption tasks (LLaMAB and QwenAB)**
>
> | Model  | Task      | Exact (%)                    | BLEU-2                        | BLEU-4                        | ROUGE-1                       | ROUGE-2                       | ROUGE-L                       | METEOR                        |
> |--------|-----------|-----------------------------|-------------------------------|-------------------------------|-------------------------------|-------------------------------|-------------------------------|-------------------------------|
> | LLaMAB | Binding   | $3.45 \pm 0.18$             | $29.01 \pm 0.36$              | $16.55 \pm 0.31$              | $37.18 \pm 0.18$              | $17.99 \pm 0.12$              | $35.94 \pm 0.21$              | $37.75 \pm 0.29$              |
> | LLaMAB | Disease   | $30.49 \pm 0.67$            | $54.84 \pm 0.90$              | $44.53 \pm 1.00$              | $62.08 \pm 0.66$              | $47.12 \pm 0.73$              | $61.12 \pm 0.66$              | $63.03 \pm 0.43$              |
> | LLaMAB | Function  | $1.06 \pm 0.07$             | $21.82 \pm 0.29$              | $10.15 \pm 0.17$              | $33.93 \pm 0.54$              | $15.53 \pm 0.38$              | $31.77 \pm 0.47$              | $33.16 \pm 0.54$              |
> | LLaMAB | Mechanism | $2.91 \pm 0.21$             | $33.15 \pm 0.21$              | $19.78 \pm 0.22$              | $44.98 \pm 0.24$              | $25.50 \pm 0.32$              | $42.59 \pm 0.29$              | $44.04 \pm 0.23$              |
> | QwenAB | Binding   | $3.78 \pm 0.38$             | $27.98 \pm 0.71$              | $16.39 \pm 0.67$              | $36.88 \pm 0.75$              | $18.00 \pm 0.80$              | $35.83 \pm 0.73$              | $36.78 \pm 0.73$              |
> | QwenAB | Disease   | $31.95 \pm 0.54$            | $56.12 \pm 0.24$              | $46.47 \pm 0.21$              | $62.70 \pm 0.45$              | $48.19 \pm 0.45$              | $61.84 \pm 0.48$              | $63.56 \pm 0.33$              |
> | QwenAB | Function  | $1.11 \pm 0.26$             | $21.05 \pm 2.83$              | $10.04 \pm 1.55$              | $32.44 \pm 4.09$              | $14.75 \pm 2.85$              | $30.44 \pm 4.11$              | $31.73 \pm 3.97$              |
> | QwenAB | Mechanism | $3.00 \pm 0.13$             | $33.01 \pm 0.34$              | $19.91 \pm 0.23$              | $45.38 \pm 0.32$              | $26.10 \pm 0.33$              | $43.17 \pm 0.36$              | $44.02 \pm 0.40$              |
>
> For clarity, we only report fold-averaged results here, and the full fold-wise tables are included in the rebuttal version paper (**Appendix G.7**).

---

> ### Author Response · Authors · 2025-11-20
> **Response Part 5**
>
> > **W6.**  Weak discussion of relation to prior instruction datasets.
>
>  We appreciate the reviewer for highlighting this point, as it allows us to elucidate the fundamental distinctions between AFD-Instruction and Evola.
>
>
> - First, Evola serves as a proteome-wide functional Q\&A model: its training data are large-scale AI-generated protein–question–answer triples derived from Swiss-Prot, TrEMBL, and ProTrek, with a focus on generic protein annotation (including Enzyme Commission numbers, Gene Ontology terms, localization information, and disease associations). In contrast, AFD-Instruction is explicitly centered around antibodies. Each example aligns a mature, fully paired antibody variable region (VH and VL, including complete CDR3) to concise, machine-readable descriptors of target antigen, disease context, binding site, mechanism, and functional activity, and further includes function-guided design instructions (e.g., generating variable regions or CDR3s conditioned on antigen and desired activity). This antibody-specific, design-oriented instruction space is not covered by Evola's general protein schema. Therefore, AFD-Instruction stands as an independent work rather than a complement to Evola.
>
> - Second, from a biological standpoint, antibody specificity is not a simple per-protein label. Antibodies are modular heterodimers: the same heavy chain can have very different specificity when paired with different light chains, and many UniProt immunoglobulin entries correspond to germline-like V/J/C segments rather than complete variable regions. Notably, the heavy-chain CDR3 is the most diverse and often the most critical determinant of the paratope; small sequence changes in CDR3 can dramatically alter antigen specificity and affinity, which is why many practical antibody engineering tasks explicitly operate at the CDR3 level. Their annotations are usually high-level (“antigen binding”, general immune function) and rarely include explicit antigen sequences, paired chains, epitope-level binding sites, or neutralization/blocking mechanisms. Evola treats such entries as ordinary proteins and generates QA pairs from the same UniProt information-point pipeline, so the supervision remains coarse and protein-entry-centric. In contrast, AFD-Instruction explicitly captures antibody-antigen-sequence relationships and downstream design behavior, enabling instructions that require reasoning about antibody–antigen specificity (e.g., “design a CDR3 that preserves neutralization of antigen X while reducing off-target binding”), which cannot be obtained simply by filtering Evola’s generic protein QA triples for immunoglobulins.
>
> - Third, to address the reviewer's concern about our insufficient discussion of prior instruction datasets, we provide a dedicated analysis in **Appendix D** (“Comparison of proteome-wide and antibody-specific supervision”). In this section, we explicitly compare proteome-wide corpora such as Evola with our focused approach to antibody-specific supervision.
>
> > **Q1.**  Can models trained on AFD-Instruction transfer to unseen antigen families or novel antibody mechanisms (e.g., pH-dependent binding, conformational epitopes)? How do they perform under cross-antigen split evaluation?
>
> - As discussed in our response to **Weakness 3**, we explicitly evaluate cross-antigen generalization by performing a family-level split based on MMseqs2 clustering at 30% sequence identity. Under this strict setting, no antigen family appears in both train and test. The AFD-tuned models (LLaMAB, QwenAB) retain strong performance across QA, captioning and design metrics and consistently outperform their untuned backbones, indicating that they do not simply memorize training antigens but can transfer to unseen antigen families.
>
> - Beyond this cross-antigen split, we also conduct a temporal out-of-distribution (OOD) evaluation using antibodies deposited in the PDB after September 2025. This OOD set covers a broader and shifted collection of indications (e.g., chronic HBV infection, mpox/orthopoxvirus infections, malaria, Alzheimer’s-related amyloid targets, immune cell homing and transplantation rejection, and various viral and cancer antigens) that are under-represented in the original SAbDab distribution. On this more challenging OOD benchmark, the AFD-tuned models show a moderate drop in absolute scores—as expected under a distribution shift—yet still substantially outperform the corresponding baseline LLMs. These results suggest that AFD-Instruction supports non-trivial transfer to both unseen antigen families and functionally distinct antibody–antigen systems. Detailed temporal OOD results and task-wise metrics are provided in **Appendix G.3** of the rebuttal version.

---

> ### Author Response · Authors · 2025-11-20
> **Response Part 6**
>
> > **Q2.**  How much do the multi-agent refinement, template expansion, and expert QC respectively improve accuracy or diversity of annotations?
>
> Please see our response to W5.
>
> > **Q3.**  Given only 4 K unique antibodies, how do you ensure the 430 K instructions are not redundant paraphrases? Did you analyze instruction uniqueness or semantic clustering?
>
> Please see our response to W2, where we provide an analysis of instruction uniqueness and semantic diversity.
>
> > **Q4.** Beyond domain restriction, what conceptual advances justify AFD-Instruction as a new dataset family rather than a subset or extension of Evola?
>
> Please see our response to W6.
>
> > **Q5.** Would it be feasible to include domain-expert evaluation of generated mechanism descriptions, or at least automated structure-text alignment (e.g., verifying residue mentions against structural contacts)?
>
> We have addressed this point in our response to W4.

---

> ### Author Response · Authors · 2025-11-25
>
> Dear Reviewer KkCC,
>
> We sincerely appreciate the time and effort you have dedicated to reviewing our manuscript and providing valuable suggestions. We submitted detailed clarifications in response to your concerns a few days ago. If you have any further feedback, questions, or issues regarding our response or the paper itself, we would be grateful to hear from you and welcome the opportunity for further discussion.
>
> Best regards,
>
> Authors

---

### Official Review · Reviewer_jopq · 2025-10-30

**Soundness:** 3
**Presentation:** 3
**Contribution:** 3
**Rating:** 4
**Confidence:** 3

**Summary:**

This work introduces the first large-scale instruction dataset specifically designed for antibodies, linking their sequences with concise functional descriptions to enable large language models to both interpret and design antibodies. By curating over 430,000 verified instruction–response pairs from structural databases and literature through a multi-agent extraction and self-questioning system, the dataset captures detailed annotations related to targets, mechanisms, and functions. Fine-tuning general LLMs such as LLaMA and Qwen on this resource produces specialized models—LLaMAB and QwenAB—that significantly outperform existing protein and general-purpose models in antibody classification, functional reasoning, and sequence generation tasks. Additionally, the models generate biologically plausible antibody sequences that demonstrate enhanced stability, binding affinity, and structural accuracy. Overall, AFD-Instruction bridges the gap between natural language guidance and molecular design, providing a new foundation for interpretable and goal-directed antibody discovery.

**Strengths:**

1 The paper introduces AFD-Instruction, the first large-scale instruction dataset for antibodies that pairs antibody sequences with structured natural language functional descriptions. This resource fills a key gap left by previous sequence-only datasets lacking functional or semantic supervision, enabling models to learn the mapping between sequence and function.

2 The authors propose a multi-agent pipeline—comprising Mr. Extractor, Dr. Mechanism, and Prof. Function—and combine self-questioning and template-based generation to produce high-quality instruction data. A rigorous quality control process involving both automated validation and expert review ensures biological reliability and factual consistency across annotations.

3 Models fine-tuned on AFD-Instruction, such as LLaMAB and QwenAB, achieve state-of-the-art results in antibody classification, functional reasoning, and design tasks. Their generated antibodies exhibit greater structural stability and stronger binding affinity compared with existing protein language models, demonstrating high potential for antibody understanding and rational antibody design in therapeutic discovery.

**Weaknesses:**

1 Although the AFD-Instruction dataset is large and comprehensive, its functional annotations mainly rely on literature-derived and database-extracted descriptions. These primarily cover common mechanisms such as neutralization, blocking, and binding-site recognition, but lack more fine-grained or dynamic functional information such as epitope escape, affinity modulation, or immune regulation.

2 The paper only validates its approach using two model families, Qwen and LLaMA, which limits the generalizability of the results across a broader range of large language model architectures. Moreover, both employed models have relatively modest parameter sizes. It is recommended to include experiments involving a wider variety of model types and parameter scales to more comprehensively evaluate the robustness and scalability of the proposed method.

3 In terms of consistency, the authors implemented a hybrid quality control protocol that combines automated semantic checks with limited expert review, but since only a small subset of data was manually verified, subtle semantic inconsistencies may persist.

**Questions:**

Same as weakness.

---

> ### Author Response · Authors · 2025-11-20
> **Response Part 1**
>
> Dear Review jopq,
>
> Thank you for your thoughtful and constructive review. We appreciate the time and care you devoted to evaluating our work, and we address your comments below.
>
> > **W1.**  Lack more fine-grained or dynamic functional information such as epitope escape, affinity modulation, or immune regulation.
>
> We appreciate the reviewer’s comment and recognize that our original presentation may not have fully conveyed the detail in the annotations. To clarify this, we manually reviewed the descriptions in AFD-Instruction and classified each entry based on whether it contained fine-grained or dynamic functional information. Our analysis reveals that **97.17%** of the descriptions include at least one of these detailed mechanisms, demonstrating that such information is widely represented throughout the dataset.
>
> Here are a few representative examples that explicitly capture these fine-grained or dynamic mechanisms:
>
> ```
> "7or9": "Steric disruption of binding due to L452R mutation", "Structural specificity for L452 without involvement of residues 478 or 484"
> ```
>
> ```
> "7jv2": "Epitope accessibility across open and closed spike conformations", "Direct competition with ACE2 for specific RBD residues (444-449, 472-498)", "Stoichiometric binding to each protomer of the spike trimer"
> ```
>
> ```
> "4ene": "Independent homodimeric subunit coordination of Cl- and H+ movement", "Proton-coupled anion transport via key glutamate residues (E148 and E203)", "Leveraging proton gradients for thermodynamically unfavorable anion transport"
> ```
>
> ```
> "7n0u": "Epitope mapping to specific residues (32–37 and 138–153)", "Coordinated heavy and light chain interactions in Fab region"
> ```
>
> These examples illustrate that our dataset contains detailed functional annotations that cover a wide range of mechanisms, providing a rich resource for teaching models about antibody function beyond just the common categories.

---

> ### Author Response · Authors · 2025-11-20
> **Response Part 2**
>
> > **W2.**  It is recommended to include experiments involving a wider variety of model types and parameter scales to more comprehensively evaluate the robustness and scalability of the proposed method.
>
> To investigate the robustness and scalability of AFD-Instruction across diverse parameter scales and model families, we also conducted instruction-tuning experiments on Gemma-9B [1], DeepSeek-MoE-16B [2], and LLaMA-70B [3]. Each model was fine-tuned on the same antibody-functional instruction dataset under identical sampling and hyperparameter settings. Across QA, caption, and design tasks, these models achieve performance that is closely comparable to QwenAB and LLaMAB, with consistently high accuracies/F1 scores and strong generation and structure metrics, indicating that the benefits of AFD-Instruction are stable across both dense and MoE architectures from 9B up to 70B parameters.
>
> **Table 1. QA Task**
>
> | Model            | Params | Class ACC/F1     | Disease ACC/F1     | Binding ACC/F1      | Mechanism ACC/F1     | Function ACC/F1       |
> |-------------------|--------|-------------------|----------------------|------------------------|------------------------|-------------------------|
> | GemmaAB          | 9.0B   | 96.19 / 95.24     | 88.51 / 84.07        | 89.11 / 89.06         | 94.14 / 93.99          | 87.96 / 87.92           |
> | DeepSeekAB-MoE   | 16B    | 98.70 / 97.89     | 86.59 / 79.35        | 88.24 / 88.22         | 94.01 / 93.85          | 86.73 / 86.71           |
> | LLaMAB-70B       | 70B    | 98.59 / 98.23     | 87.11 / 82.25        | 88.35 / 88.29         | 93.69 / 93.53          | 87.45 / 87.41           |
>
>
> **Table 2. Caption Task**
>
> | Model            | Params | Domain     | EM   | BLEU-2 | BLEU-4 | ROUGE-1 | ROUGE-2 | ROUGE-L | METEOR |
> |-------------------|--------|------------|------|---------|--------|----------|----------|----------|--------|
> | GemmaAB          | 9.0B   | Binding    | 5.64 | 32.09   | 19.72  | 41.74    | 22.36    | 40.40    | 41.44 |
> | DeepSeekAB-MoE   | 16B    | Binding    | 5.71 | 36.74   | 24.25  | 48.55    | 29.71    | 46.26    | 47.55 |
> | LLaMAB-70B       | 70B    | Binding    | 5.78 | 33.24   | 19.89  | 43.50    | 23.13    | 42.21    | 42.83 |
> | GemmaAB          | 9.0B   | Disease    | 35.66| 59.01   | 49.26  | 66.48    | 52.06    | 65.60    | 66.95 |
> | DeepSeekAB-MoE   | 16B    | Disease    | 34.17| 57.58   | 47.91  | 64.26    | 50.08    | 63.43    | 64.99 |
> | LLaMAB-70B       | 70B    | Disease    | 34.74| 60.01   | 49.24  | 68.24    | 52.50    | 67.26    | 68.70 |
> | GemmaAB          | 9.0B   | Mechanism   | 4.55 | 35.68   | 22.71  | 48.03    | 28.88    | 45.80    | 46.66 |
> | DeepSeekAB-MoE   | 16B    | Mechanism   | 4.11 | 34.71   | 21.82  | 46.85    | 28.00    | 44.69    | 45.75 |
> | LLaMAB-70B       | 70B    | Mechanism   | 5.92 | 38.58   | 25.46  | 50.98    | 31.20    | 48.57    | 49.93 |
> | GemmaAB          | 9.0B   | Function     | 1.21 | 24.26   | 12.39  | 36.88    | 18.24    | 34.49    | 35.35 |
> | DeepSeekAB-MoE   | 16B    | Function     | 1.49 | 23.46   | 11.82  | 35.70    | 17.69    | 33.60    | 34.39 |
> | LLaMAB-70B       | 70B    | Function     | 1.46 | 24.03   | 11.93  | 37.06    | 17.84    | 34.82    | 35.90 |
>
>
> **Table 3. Design Task**
>
> | Model            | Perplexity ↓ | SRR ↑ | ipTM ↑ | pTM ↑ | pLDDT ↑ |
> |------------------|--------------|-------|--------|-------|---------|
> | **GemmaAB**       | 2.932        | 0.872 | 0.468  | 0.739 | 0.879   |
> | **DeepSeekAB-MoE**| 2.901        | 0.878 | 0.472  | 0.747 | 0.880   |
> | **LLaMAB-70B**    | 2.871        | 0.883 | 0.491  | 0.759 | 0.883   |
>
>
> [1] Team, Gemma, et al. "Gemma 2: Improving open language models at a practical size." arXiv preprint arXiv:2408.00118 (2024).
>
> [2] Dai, Damai, et al. "Deepseekmoe: Towards ultimate expert specialization in mixture-of-experts language models." arXiv preprint arXiv:2401.06066 (2024).
>
> [3] Dubey, Abhimanyu, et al. "The llama 3 herd of models." arXiv e-prints (2024): arXiv-2407.

---

> ### Author Response · Authors · 2025-11-20
> **Response Part 3**
>
> > **W3.**  In terms of consistency, the authors implemented a hybrid quality control protocol that combines automated semantic checks with limited expert review, but since only a small subset of data was manually verified, subtle semantic inconsistencies may persist.
>
> We agree that large-scale automatically constructed instruction datasets inevitably involve a trade-off between coverage and perfect semantic consistency. This limitation is inherent to any large synthetic corpus generated at scale, *no such dataset can guarantee complete semantic uniformity*, since only a subset can be manually examined in practice.
>
> To mitigate this, we implemented a substantially stronger quality-control pipeline than previous works. Our protocol combines template-level sanity constraints, multi-agent self-correction, and targeted expert review to calibrate templates and remove systematic errors. Although these procedures cannot eliminate every subtle inconsistency, they substantially improve annotation reliability and reduce noise propagation.
>
> Importantly, the effectiveness of this layered QC process is reflected in downstream performance: models trained on AFD-Instruction consistently achieve higher accuracy and robustness than strong baseline LLMs across all evaluated tasks, indicating that the remaining noise does not hinder practical learning. Consistent with this, our ablation study (**Appendix~G.5 in the rebuttal version paper**) shows clear performance degradation whenever the multi-agent refinement or expert QC components are removed, further supporting the contribution of our QC design. The ablation results of the QC are as follows:
>
> **Table 1. Ablation study on antibody sequence understanding (classification)**
>
> | Variant         | Class ACC | Class F1 | Disease ACC | Disease F1 | Binding ACC | Binding F1 | Mechanism ACC | Mechanism F1 | Function ACC | Function F1 |
> |:----------------|----------:|---------:|------------:|-----------:|------------:|-----------:|--------------:|-------------:|-------------:|------------:|
> | w/o Expert QC   | 96.22     | 94.41    | 69.33       | 65.48      | 80.08       | 77.45      | 86.27         | 86.15        | 78.59        | 77.51       |
> | **Full pipeline** | **98.48** | **96.23** | **85.11**   | **80.91**  | **87.01**   | **86.96**  | **92.91**     | **92.76**    | **83.81**    | **83.74**   |
>
> ---
>
> **Table 2. Ablation study on antibody sequence understanding (caption)**
>
> | Domain    | Variant         | Exact | BLEU-2 | BLEU-4 | ROUGE-1 | ROUGE-2 | ROUGE-L | METEOR |
> |:----------|:----------------|------:|------:|------:|--------:|--------:|--------:|------:|
> | Binding   | w/o Expert QC   | 2.93  | 27.50 | 15.80 | 37.28   | 17.80   | 30.46   | 37.28 |
> | Binding   | **Full pipeline** | **4.10** | **29.78** | **17.05** | **38.49** | **18.68** | **37.33** | **38.84** |
> | Disease   | w/o Expert QC   | 28.01 | 51.23 | 41.11 | 61.00   | 45.47   | 60.51   | 61.19 |
> | Disease   | **Full pipeline** | **30.64** | **54.22** | **43.70** | **62.40** | **47.25** | **61.46** | **63.48** |
> | Mechanism | w/o Expert QC   | 2.31  | 31.61 | 17.79 | 43.80   | 24.13   | 41.51   | 42.54 |
> | Mechanism | **Full pipeline** | **2.83** | **33.33** | **19.68** | **45.02** | **25.30** | **42.62** | **44.26** |
> | Function  | w/o Expert QC   | 0.60  | 18.82 | 8.92  | 31.58   | 14.28   | 29.81   | 29.67 |
> | Function  | **Full pipeline** | **0.74** | **21.56** | **10.13** | **33.86** | **15.57** | **31.62** | **33.18** |
>
> ---
>
> **Table 3. Ablation study on antibody design**
>
> | Variant         | Perplexity ↓ | SRR ↑ | ipTM ↑ | pTM ↑ | pLDDT ↑ |
> |:----------------|-------------:|------:|-------:|------:|--------:|
> | w/o Expert QC   | 2.947        | 0.835 | 0.421  | 0.705 | 0.861   |
> | **Full pipeline** | **2.857**    | **0.884** | **0.476** | **0.744** | **0.880** |

---

> ### Author Response · Authors · 2025-11-25
>
> Dear Reviewer jopq,
>
> We sincerely appreciate the time and effort you have dedicated to reviewing our manuscript and providing valuable suggestions. We submitted detailed clarifications in response to your concerns a few days ago. If you have any further feedback, questions, or issues regarding our response or the paper itself, we would be grateful to hear from you and welcome the opportunity for further discussion.
>
> Best regards,
>
> Authors

---

> > ### Comment · Reviewer_jopq · 2025-11-28
> >
> > Thank you for your hard work. Your additional explanations and experiments answer my concerns to some extend. I'd like to give you more points but the system seems to be locked due to the recent "affair". If the system reopens, contact me for the score modification.

---

> > > ### Author Response · Authors · 2025-11-28
> > >
> > > Thank you very much for your time and constructive feedback. We are glad that our additional explanations and experiments have addressed your concerns, and we sincerely appreciate your willingness to increasing the score.

---

### Official Review · Reviewer_9Rm4 · 2025-11-01

**Soundness:** 3
**Presentation:** 3
**Contribution:** 2
**Rating:** 4
**Confidence:** 4

**Summary:**

The paper addresses a key challenge in bioinformatics: while Large Language Models (LLMs) are powerful, their ability to interpret and design antibodies using natural language instructions is limited. To solve this, the authors introduce AFD-Instruction, the first large-scale instruction dataset specifically for antibodies, which links sequences to detailed functional annotations. The authors fine-tuned general LLMs (Qwen2-7B and LLaMA-8B) on AFD-Instruction to create new models named QwenAB and LLaMAB. In understanding tasks, these models achieved SOTA results, significantly outperforming both general-purpose LLMs. In design tasks, the models generated antibody sequences with greater plausibility (lower perplexity) and higher fidelity to natural sequences (higher SRR).

**Strengths:**

1. While instruction tuning for general proteins exists (e.g., InstructProtein, Mol-Instructions) , this paper correctly identifies antibodies as a unique and challenging class that existing resources do not adequately cover.

**Weaknesses:**

1. The models (QwenAB, LLaMAB) show massive performance gains on classification tasks (e.g., in Table 1, 87.81% on Binding for QwenAB vs. 46.59% for GPT-4o). This dramatic improvement suggests the model might be learning spurious correlations or "template-fitting" rather than generalizable biological reasoning. The Understanding evaluation needs a more challenging, held-out test set to prove generalization.
2. The dataset is built by sampling 4,305 antibody entries from PDB and SAbDab. These databases are inherently biased towards structurally-characterized and crystallized antibodies. The paper mentions using MMseqs2 to mitigate data imbalance, but this addresses sequence redundancy, not the foundational structural and functional bias of the source data.
3. Table 3 compares models on metrics like Perplexity, SRR, and various structural scores. The AFD-tuned models (LLaMAB) perform well. However, this experiment compares models trained on different data and for different tasks. It shows that LLaMAB is a "good" antibody sequence generator in general, but it does not prove that it successfully followed the natural language functional instruction.

**Questions:**

1. Could you evaluate your model's performance on a temporal hold-out set (i.e., antibodies and papers published after your data collection cutoff)? This would provide much stronger evidence of true generalization.

---

> ### Author Response · Authors · 2025-11-20
> **Response Part 1**
>
> Dear Review 9Rm4,
>
> Firstly, we express our gratitude for your thorough review and insightful comments on our paper. We have taken your feedback as an opportunity to further refine our manuscript. We address your concerns below:
>
>
> > **W1.** The Understanding evaluation needs a more challenging, held-out test set to prove generalization.
>
> In response, we have conducted a more rigorous evaluation using a temporally held-out test set, which was designed to test generalization beyond the original dataset.
>
> Specifically, we construct a new out-of-distribution (OOD) test set by collecting 33 antibodies that were released after September 2025 and are supported by primary literature. For each antibody, we manually read the associated papers, extract key functional and mechanistic information, and design corresponding question–answer pairs. This yields a strictly time-disjoint OOD benchmark whose annotations are directly grounded in experimental studies. The basic information for these 33 antibodies is provided in **Appendix G.3 (Table 8)** of the rebuttal version of the paper.
>
> We then evaluated both QwenAB and LLaMAB on this OOD antibody test set for two task formats: caption-style and QA-style. The results, summarized in the tables below, show that while there is some performance degradation relative to the original in-distribution test set, the models still exhibit robust performance on previously unseen, literature-derived antibodies, indicating that the performance improvements reported in Table 1 are not merely due to spurious correlations or template fitting.
>
> **Table 1. LLaMAB Performance on the Temporal OOD Antibody Test Set**
>
> | Domain    | EM   | B-2   | B-4   | R-1   | R-2   | RL    | MET   | ACC(QA) | F1(QA)  |
> |:----------|-----:|------:|------:|------:|------:|------:|------:|--------:|--------:|
> | Binding   | 3.67 | 23.69 | 15.86 | 36.18 | 17.48 | 34.71 | 34.44 | 86.21   | 85.12   |
> | Disease   | 28.32| 50.33 | 39.27 | 59.47 | 45.30 | 58.45 | 60.20 | 83.23   | 72.21   |
> | Function  | 0.63 | 20.32 | 9.73  | 30.13 | 13.20 | 31.12 | 30.29 | 82.72   | 82.25   |
> | Mechanism | 2.73 | 31.25 | 18.72 | 39.76 | 23.72 | 41.03 | 42.56 | 89.61   | 88.82   |
> | Class     |   |    |    |    |    |    |   | 97.23   | 96.14   |
>
> **Table 2. QwenAB Performance on the Temporal OOD Antibody Test Set**
>
> | Domain    | EM   | B-2   | B-4   | R-1   | R-2   | RL    | MET   | ACC(QA) | F1(QA)  |
> |:----------|-----:|------:|------:|------:|------:|------:|------:|--------:|--------:|
> | Binding   | 4.13 | 22.12 | 16.42 | 34.95 | 18.02 | 33.24 | 34.01 | 86.32   | 86.04   |
> | Disease   | 30.12| 53.29 | 40.12 | 58.14 | 46.12 | 58.72 | 61.27 | 85.31   | 81.61   |
> | Function  | 0.68 | 21.02 | 10.08 | 32.15 | 14.89 | 31.73 | 31.52 | 83.75   | 82.74   |
> | Mechanism | 2.03 | 28.14 | 17.28 | 40.13 | 22.27 | 39.56 | 40.50 | 91.63   | 90.78   |
> | Class     |   |    |   |  |   |   |   | 98.01   | 96.45   |

---

> ### Author Response · Authors · 2025-11-20
> **Response Part 2**
>
> > **W2.** The paper mentions using MMseqs2 to mitigate data imbalance, but this addresses sequence redundancy, not the foundational structural and functional bias of the source data.
>
> We fully acknowledge this issue and would like to clarify how this limitation relates to the goals of our work.
>
>
> **(1) Structural and Functional Bias in Antibody Datasets:**
>
> The bias identified is a well-known characteristic of all structure-based antibody datasets, not a limitation unique to AFD-Instruction. Antibodies can only enter PDB/SAbDab after undergoing successful expression, purification, crystallization/cryo-EM imaging, and mechanistic characterization. As a result, these databases naturally over-represent antibodies that are experimentally tractable, clinically relevant, or of high research interest.
>
> **(2) Why Structural Characterization is Essential for Functional Annotation:**
>
> Structural characterization enables reliable functional annotation, as critical properties like binding site, epitope class, neutralization mechanism, blocking activity, and allosteric effects cannot be inferred from raw sequences alone. These insights come from structural studies, which is why existing functional antibody knowledge is overwhelmingly concentrated in PDB and SAbDab-linked literature. To build a function-aligned instruction dataset, using these sources is necessary.
>
> **(3) Focus on Functional Alignment:**
>
> Our goal is functional alignment, not unbiased modeling of the entire antibody sequence universe. While some databases contain millions of immune repertoire sequences, they often lack functional annotations. In contrast, PDB and SAbDab offer high-quality, experimentally validated functional descriptions, making them essential for training models that interpret and generate function-conditioned antibody behavior.
>
> **(4) Functional Bias:**
>
> Regarding functional bias (e.g., enrichment of HIV and SARS-CoV-2), we acknowledge this reflects a limitation of global research landscape.
>
> - Antibodies that enter SAbDab are typically high-affinity, optimized, and mechanistically well studied. They cluster around therapeutically important antigens, which is why the structural literature is to enable instruction generation. While this distribution is not uniform, it represents the entirety of experimentally validated antibody functional knowledge available to date.
>
> - As described in our response to Weakness 1, we constructed a strict temporal OOD antibody test set. Building on this setup, we further examine the reviewer’s concern about functional bias using newly released antibodies. These antibodies target a markedly different set of antigens compared with the canonical SAbDab distribution. Specifically, the OOD set includes antibodies against: Chronic hepatitis B virus (HBV), Mpox / Orthopoxvirus infections, Malaria (Plasmodium falciparum), Alzheimer’s disease–related amyloid targets, Immune cell homing/retention mechanisms, Bacterial infections, Cancer-associated antigens, Immune rejection in allogeneic transplantation, Herpesviridae members (HSV-1, EBV, VZV), Respiratory syncytial virus A (RSV-A), Influenza. Notably, only two antibodies in this OOD set target COVID-19, which is the frequent antigen family in SAbDab. This distribution therefore provides a realistic and stringent test of functional generalization beyond the dominant clusters in the training data. The results demonstrates that the models capture transferable functional patterns rather than merely memorizing frequent antigen families in SAbDab.
>
>
> > **W3.** LLaMAB is a "good" antibody sequence generator in general, but it does not prove that it successfully followed the natural language functional instruction.
>
> To address this concern, we have added a new case study in **Appendix G.4 (Figure 13)** of the rebuttal version. In this case study, we explicitly control the functional instruction while holding all other factors constant. For a set of representative antigens, we fix the same antibody scaffold and antigen context, and generate sequences under different functional directives. Specifically, we vary the epitope focus, neutralization mechanism, or effector-related properties, while using identical sampling and scoring pipelines.
>
> The results of these controlled experiments show systematic changes in the generated antibody sequences and structures that directly correspond to the changes in the functional instruction. For instance, designs generated under different functional prompts concentrate paratope residues on distinct epitope regions and exhibit different physicochemical and mechanistic profiles that align with the textual specification. These qualitative observations confirm that the model is not only capable of generating high-quality antibody sequences, but also responds in a meaningful and interpretable way to changes in natural language functional instructions.

---

> ### Author Response · Authors · 2025-11-20
> **Response Part 3**
>
> > **Q1.** Could you evaluate your model's performance on a temporal hold-out set (i.e., antibodies and papers published after your data collection cutoff)? This would provide much stronger evidence of true generalization.
>
> Please refer to our response under **Weakness 1** (temporal hold-out evaluation) where we describe the new OOD antibody benchmark and present results demonstrating generalization.

---

> ### Author Response · Authors · 2025-11-25
>
> Dear Reviewer 9Rm4,
>
> We sincerely appreciate the time and effort you have dedicated to reviewing our manuscript and providing valuable suggestions. We submitted detailed clarifications in response to your concerns a few days ago. If you have any further feedback, questions, or issues regarding our response or the paper itself, we would be grateful to hear from you and welcome the opportunity for further discussion.
>
> Best regards,
>
> Authors

---

### Author Response · Authors · 2025-12-02
**Summary of our rebuttal for AC**

Dear AC,

Thank you again for all your efforts in handling our submission, especially given the recent unexpected circumstances.

Here is a brief, point-by-point summary in support of our case:

- **Reviewer KkCC**: They were **overall positive** and indicated that, if we clarified the outstanding issues, they would be **willing to increase their score**. We addressed each point in detail and incorporated the requested clarifications and analyses.
- **Reviewer jopq**: Their concerns mainly reflected misunderstandings and requests for additional evidence. We clarified these points and added the requested experiments; following our rebuttal, they indicated they would **increase their score**.
- **Reviewer 9Rm4**: Their primary concern was whether our gains reflect template fitting rather than generalizable reasoning. We addressed this directly with a strict temporal out-of-distribution evaluation.

In short, the concerns raised in review have been addressed concretely in the rebuttal. For clarity, we summarize the main issues and how they were resolved below:

- **Generalization versus template fitting**: We added a **strict temporal out-of-distribution benchmark** constructed from newly released antibodies supported by primary literature, and evaluated both models on it. The models retain strong performance under distribution shift, providing direct evidence that the reported gains are not driven by spurious correlations or template memorization.

- **Evidence of instruction following**: To move beyond “good generation” claims, we added a **controlled case study** that fixes the antibody scaffold and antigen context while varying only the functional directive. The resulting antibodies exhibit systematic and directive-consistent changes in sequence and structure, supporting that the models respond meaningfully to functional instructions.

- **Structural and functional bias in PDB and SAbDab sources**: We clarified that our dataset targets **antibody functional descriptions**, where key properties such as **epitope and paratope localization** and **neutralization or blocking mechanisms** can be reliably determined only after folding and forming a **three-dimensional complex**. Consequently, using **PDB and SAbDab linked structural studies** is necessary for accurate supervision. We further used the temporal out-of-distribution set, which covers a shifted set of antigens and indications, to test robustness beyond dominant clusters.

- **Scope and robustness of experimental validation**: We expanded the experiments to address requests for broader validation. This includes evaluation across **multiple model families and parameter scales**, additional **distribution-shift** settings, and systematic **ablations** that quantify the contribution of major components in the data-generation and quality-control pipeline.

- **Annotation quality and consistency**: We acknowledged that large-scale instruction corpora involve an inherent tradeoff between coverage and perfect semantic uniformity, and presented **ablation evidence** showing that **multi-stage refinement** and **expert quality control** materially improve downstream performance. This supports the reliability of the pipeline and limits the impact of residual noise.

- **Redundancy and linguistic inflation**: We provided quantitative analyses of **semantic diversity** using embedding-based clustering and nearest-neighbor similarity. The results show a broad instruction space and do not support the hypothesis that the corpus is dominated by near-duplicate paraphrases.

- **Scientific correctness beyond lexical metrics**: We complemented **BLEU and ROUGE** with **expert assessment** of generated mechanism descriptions along dimensions of **mechanistic accuracy**, **structural consistency**, and **mechanistic granularity**, providing evidence that the improvements extend beyond surface-level text similarity.

Given our clear and substantive responses to the reviewers’ concerns, as well as the improved clarity and strengthened empirical support, we believe the revised submission is significantly stronger and provides you with a clearer basis for justifying a recommendation.

---

### Note · Authors · 2026-04-18

I have read and agree with the venue's withdrawal policy on behalf of myself and my co-authors.

---

### Meta-Review · Area_Chair_ddbP · 2025-12-04

**Summary:**

The paper addresses a key challenge in bioinformatics: while Large Language Models (LLMs) are powerful, their ability to interpret and design antibodies using natural language instructions is limited. To solve this, the authors introduce AFD-Instruction, the first large-scale instruction dataset specifically for antibodies, which links sequences to detailed functional annotations.

**Reviewer Concerns:**

Most reviewers have concerns on the experiments, while the authors give detailed responses. I think these responses addressed the concerns.

**Reviewer Scores:**

One reviewer has claimed he will increase his score.

---

### Decision · Program_Chairs · 2026-01-26

Accept (Poster)